# Norm-count Hypothesis: On the Relationship Between Norm and Object Count in Visual Representations

## Abstract

We present a novel hypothesis on norms of representations produced by convolutional neural networks (CNNs). In particular, we propose the norm-count hypothesis (NCH), which states that there is a monotonically increasing relationship between the number of certain objects in the image, and the norm of the corresponding representation. We formalize and prove our hypothesis in a controlled setting, showing that the NCH is true for linear and batch normalized CNNs followed by global average pooling, when they are applied to a certain class of images. Further, we present experimental evidence that corroborates our hypothesis for CNN-based representations. Our experiments are conducted with several real-world image datasets, in both supervised and self-supervised learning – providing new insight on the relationship between object counts and representation norms.

## 1 Introduction

The ability to learn high-quality representations from a wide range of complex data types lies at the heart of the success of deep learning. Recently, several works have studied how deep learning-based representations can be embedded in non-Euclidean spaces, to further improve representation quality (Bronstein et al., 2017). In particular, embedding representations on the hypersphere using $L_2$ normalization has proven to be a particularly promising direction for several downstream applications, such classification and regression (Mettes et al., 2019; Scott et al., 2021; Tan et al., 2022), and self-supervised learning (SSL) (Chen et al., 2020; Caron et al., 2021).

However, despite the widespread use of $L_2$ normalization in several aspects of deep learning, little work exists on understanding exactly what type of information the norm contains, and why discarding this information can improve representation quality. Hence, we still lack critical understanding of the role of representation norms, and what types information they encode. In this work, we aim to improve the understanding of norms of image representations produced by convolutional neural network (CNN). Our work is built on a novel hypothesis on the relationship between CNN-based representations and the number of objects in the input image:

**Informal Definition 1** (Norm-count hypothesis)**.** There is a monotonically increasing relationship between the norm of a representation produced by a CNN, and the number of objects in the input image, for which the CNN is trained to recognize.

The norm-count hypothesis (NCH) thus proposes a theory on the relationship between norm, and the number of detections produced by the CNN. This relationship is *monotonically increasing*, meaning that increasing the number of objects in an image, will increase the norm of the image representation. In addition, we will see later that the NCH further implies that angles encode information about the types of objects detected by the CNN in the given image.

Presuming that angles encode semantic information and norms encode count information, it is not surprising that models trained to perform inherently count-invariant tasks will perform better when count information is discarded, for instance by $L_2$ normalization. Interestingly, count-invariant tasks are widely studied in

computer vision. Single-label supervised classification (*e.g.* ImageNet (Deng et al., 2009)), whole-image clustering, linear evaluation in self-supervised learning, and standard evaluation setups in few-shot learning all belong to the category of count-invariant tasks.

The objective of our work is to formalize and experimentally evaluate the NCH for CNNs in supervised and self-supervised learning. Our main contributions are:

1. We propose the NCH – stating that there is a monotonically increasing relationship between the norm of a representation, and the number of objects in the given image.

2. We prove that the NCH is true in a controlled setting, assuming that input images are composed of several *object images*, for which the feature extractor provides a delta-like response in a single channel.

3. We conduct an extensive experimental evaluation with synthetic and real-world datasets, with both supervised and self-supervised models. Our results show monotonically increasing relationships between norm and count for the majority of models and datasets – corroborating the NCH.

The rest of the paper is structured as follows: Section 2 gives an overview of work related to ours. In Section 3, we theoretically analyze the NCH, and prove that it holds under certain assumptions. Section 4 includes the results of our experiments. We finish the paper with Section 5, presenting some concluding remarks and directions for future work.

## 2 Related work

In this section, we summarize other work related to this paper. We emphasize that our work is complementary to these, as none of the works below provide an accurate and rigorous understanding of the information contained in norms of CNN-based representations.

### 2.1 Embedding representations on the hypersphere

Embedding representations on the hypersphere instead of in Euclidean space has shown to be beneficial for both supervised classification and regression (Mettes et al., 2019; Scott et al., 2021; Tan et al., 2022). Mettes et al. (2019) develop classification and regression losses on the hypersphere, illustrating that $L_2$ normalized representations and prototypes are beneficial for both classification and regression. The more recent work by Tan et al. (2022) shows that a supervised classification model can be regularized with a self-supervised contrastive loss on the unit hypersphere.

$L_2$ normalization is also common in models for self-supervised learning of image representations (Chen et al., 2020; He et al., 2020; Grill et al., 2020; Caron et al., 2020; 2021; Goyal et al., 2022; Li et al., 2023). The benefit of $L_2$ normalization appears to stem from similarity measures and contrastive losses being more well-behaved after discarding the norm – resulting in compact and well-separated classes (Wang & Isola, 2020).

### 2.2 Hyperspherical regularization

Hyperspherical embedding of network weights have also shown to be beneficial to regularize training of deep neural networks (DNNs) (Salimans & Kingma, 2016; Liu et al., 2017; 2018; 2021). These methods constrain the weight vectors in DNNs to lie on the unit hypersphere. Liu et al. (2017) show that hyperspherical weights improve the conditioning of the optimization problem, helping the optimizer converge faster to potentially better solutions. However, our work is orthogonal to this, since we aim to understand norms of *representations*, and not norms of weights.

## 3 Norm-count hypothesis

The purpose of this section is to formalize the NCH, and to analyze it in a rigorous theoretical setting. To do so, we assume certain properties of the feature extractor (*e.g.* CNN). These feature extractors admit a certain class of images, referred to as *object images*, which can be seen as "prototypes" of the objects the feature extractor is trained to detect.

Having established properties of the feature extractor and corresponding object images, we prove that the norm of the representation produced by the feature extractor is proportional to the number of object images present in the given image. Thereby corroborating the NCH. Proofs for the results presented in this section are given in Appendix A.

We start by providing exact definitions of images and image translation.

**Definition 1** (Images)**.** The set of images with $C$ channels and size $W \times H$ is defined as

$$\mathcal{I}_{C,W,H} = \{I : \mathbb{N}^0_{<C} \times \mathbb{Z} \times \mathbb{Z} \to \mathbb{R} \mid I(c,x,y) = 0 \text{ if } (x,y) \notin \mathbb{N}^0_{<W} \times \mathbb{N}^0_{<H}\} \tag{1}$$

where $\mathbb{N}^0_{<a} = \{0, 1, \dots, a-1\}$.

**Definition 2** (Translation operator)**.** A translation operator $\mathrm{Tr}_{x',y'} : \mathcal{I}_{C,W,H} \to \mathcal{I}_{C,W,H}$ shifts the given image by $x', y'$ pixels

$$\mathrm{Tr}_{x',y'}(I)(c,x,y) = I(c, x-x', y-y') \tag{2}$$

**Definition 3** (Translation equivariance)**.** A mapping $f : \mathcal{I}_{C,W,H} \to \mathcal{I}_{C',W',H'}$ is translation equivariant iff for a translation operator $\mathrm{Tr}_{x',y'}$, we have

$$f \circ \mathrm{Tr}_{x',y'} = \mathrm{Tr}_{x',y'} \circ f \tag{3}$$

where $\circ$ denotes the composition of functions.

We will now define two types of functionals acting on images, namely *strict detectors* and *relaxed detectors*. Detectors are generic functionals with certain constraints on how they act on element-wise sums of images. As we will show later, detectors are closely related to CNNs, and when combined with a pooling operator (defined later), detectors are responsible for producing vectorial representations for a given image.

**Definition 4** (Strict detector)**.** A strict detector is a translation invariant mapping $f : \mathcal{I}_{C,W,H} \to \mathcal{I}_{C',W',H'}$ that satisfies

$$f(I_1 + I_2) = f(I_1) + f(I_2) + A_f \tag{4}$$

where $A_f \in \mathcal{I}_{C',W',H'}$ is a constant image independent of $I_1$ and $I_2$, satisfying $A_f(k,x,y) = a_{f,k}$ for all $(k,x,y) \in \mathbb{N}^0_{<C'} \times \mathbb{N}^0_{<W'} \times \mathbb{N}^0_{<H'}$. Addition is defined element-wise.

**Definition 5** (Relaxed detector)**.** A relaxed detector is a translation invariant mapping $f : \mathcal{I}_{C,W,H} \to \mathcal{I}_{C',W',H'}$ that satisfies

$$|f(I_1 + I_2)| \preccurlyeq |f(I_1)| + |f(I_2)| + |A_f| \tag{5}$$

where $A_f \in \mathcal{I}_{C',W',H'}$ is a constant image independent of $I_1$ and $I_2$, satisfying $A_f(k,x,y) = a_{f,k}$ for all $(k,x,y) \in \mathbb{N}^0_{<C'} \times \mathbb{N}^0_{<W'} \times \mathbb{N}^0_{<H'}$. Addition and absolute value are defined element-wise, and $I_1 \preccurlyeq I_2$ implies that $I_1(k,x,y) \leq I_2(k,x,y)$ for all $k, x, y$.

The following propositions show properties of detectors that are relevant for CNNs.

**Proposition 1** (Composition of detectors)**.** *For detectors $f$ and $g$, the following holds:*

1. *If $f$ and $g$ are strict detectors, then $g \circ f$ is a strict detector with $A_{g \circ f} = g(A_f) + 2A_g$*

2. *If $f$ is a strict detector and $g$ is a relaxed detector, then $g \circ f$ is a relaxed detector with $A_{g \circ f} = |g(A_f)| + 2|A_g|$.*

**Proposition 2** (Convolutions are strict detectors.)**.** *Let $\mathrm{Conv}_K : \mathcal{I}_{C,W,H} \to \mathcal{I}_{1,W+w-1,H+h-1}$ be the convolution operator convolving the given image, $I \in \mathcal{I}_{C,W,H}$, with a filter, $K \in \mathcal{I}_{C,h,w}$*

$$\mathrm{Conv}_K(I)(0,x,y) = \sum_{c=0}^{C-1} \sum_{x'=-\infty}^{\infty} \sum_{y'=-\infty}^{\infty} K(c,x',y')I(c,x-x',y-y'). \tag{6}$$

*Then $\mathrm{Conv}_K$ is a strict detector with $(C',H',W') = (1, W+w-1, H+h-1)$ and $A_{\mathrm{Conv}_K} = 0$.*

Since images are infinitely zero-padded (Definition 1), Proposition 2 assumes that "full" padding is used in the convolution operator. However, we note that the proposition also holds if "valid" or "same" padding is used.

**Proposition 3** (Batch-normalization in inference mode is a strict detector)**.** *Let $\mathrm{BN}_{\mathbf{b},\mathbf{g},\boldsymbol{\mu},\boldsymbol{\sigma}} : \mathcal{I}_{C,W,H} \to \mathcal{I}_{C,W,H}$ represent batch normalization in inference mode, with parameters $\mathbf{b} = [b_0,\ldots,b_{C-1}]^\top, \mathbf{g} = [g_0,\ldots,g_{C'-1}]^\top$ and running moment estimates $\boldsymbol{\mu} = [\mu_0,\ldots,\mu_{C-1}]^\top, \boldsymbol{\sigma} = [\sigma_0,\ldots,\sigma_{C'-1}]^\top$, defined as*

$$\mathrm{BN}_{\mathbf{b},\mathbf{g},\boldsymbol{\mu},\boldsymbol{\sigma}}(I)(k,x,y) = \frac{I(k,x,y) - \mu_k}{\sigma_k} g_k + b_k. \tag{7}$$

*Then $\mathrm{BN}_{\mathbf{b},\mathbf{g},\boldsymbol{\mu},\boldsymbol{\sigma}}$ is a strict detector with $(C',W',H') = (C,W,H)$ and $a_{\mathrm{BN}_{\mathbf{b},\mathbf{g},\boldsymbol{\mu},\boldsymbol{\sigma}},k} = \frac{\mu_k}{\sigma_k} g_k - b_k$.*

**Proposition 4** (LeakyReLU is a relaxed detector.)**.** *Let $\mathrm{LeakyReLU}_\alpha : \mathcal{I}_{C,W,H} \to \mathcal{I}_{C,W,H}$ be defined element-wise as*

$$\mathrm{LeakyReLU}_\alpha(I(k,x,y)) = \begin{cases} I(k,x,y), & \text{if } I(k,x,y) > 0 \\ \alpha \cdot I(k,x,y), & \text{otherwise} \end{cases} \tag{8}$$

*for all $k,x,y$, and $\alpha \in [0,1)$. Then $\mathrm{LeakyReLU}_\alpha$ is a relaxed detector with $(C',W',H') = (C,W,H)$ and $A_{\mathrm{LeakyReLU}_\alpha} = 0$.*
*This also holds for the standard $\mathrm{ReLU}(x) = \max\{0,x\}$ activation, since $\mathrm{ReLU} = \mathrm{LeakyReLU}_0$.*

From Propositions 1, 2 and 3, we see that linear and batch normalized CNNs – *i.e.* networks consisting only of compositions of convolutions and batch normalization – are strict detectors. Furthermore, combining Propositions 1, 2 and 4 shows that a CNN consisting of convolutions and LeakyReLU (or ReLU) activations are compositions of relaxed detectors. These propositions thus cover the most important building blocks of CNNs, along with the most common activation functions.

CNNs for classification and representation learning are often followed by a global pooling operator that aggregates information from all spatial locations. The following definition considers a general pooling operation, which we use in our theoretical analysis. We then prove that global average pooling (GAP) – one of the most frequently used pooling operations – is a special case of the general pooling operation.

**Definition 6** (Global pooling operator)**.** Let $I \in \mathcal{I}_{C,W,H}$ be an image. The mapping $\mathrm{Pool} : \mathcal{I}_{C,W,H} \to \mathbb{R}^C$ is called a global pooling operator if there exists non-negative real numbers $\gamma_0,\ldots,\gamma_{C-1}$ independent of $I$, such that

$$|\mathrm{Pool}(I)_k| \le \gamma_k \left| \sum_{x=0}^{W-1} \sum_{y=0}^{H-1} I(k,x,y) \right|, \quad k \in \mathbb{N}^0_{<C}. \tag{9}$$

**Proposition 5** (GAP is a global pooling operator). *For an image $I \in \mathcal{I}_{C,W,H}$, let GAP be defined as*

$$\text{GAP}(I)_k = \frac{1}{WH} \sum_{x=0}^{W-1} \sum_{y=0}^{H-1} I(k,x,y), \quad k \in \mathbb{N}^0_{<C}. \tag{10}$$

*Then* GAP *is a global pooling operator with $\gamma_k = \frac{1}{WH}$, $\forall k \in \mathbb{N}^0_{<C}$.*

Our objective is now to understand norms of representations computed by a detector followed by a global pooling operator. Hence, we define object images as "prototypical" images that give a delta-like response when processed by the given detector.

**Definition 7** (Object images). An image $O_j \in \mathcal{I}_{C,h,w}$ is said to be an object image of type $j$ w.r.t. the detector $f$, iff

$$f(O_j) = \delta_{j,0,0} \tag{11}$$

where $\delta_{j,x',y'}$ is the Kronecker delta function

$$\delta_{j,x',y'}(k,x,y) = \begin{cases} 1, & (k,x,y) = (j,x',y') \\ 0, & \text{otherwise} \end{cases}. \tag{12}$$

The set of all object images of type $j$ is denoted $\Omega_j = \{O \mid f(O) = \delta_{j,0,0}\}$.

In a highly accurate supervised model the object image types will tend to coincide with the classes the model is trained to detect. This is because supervised models are trained to output a one-hot prediction vector, resembling the delta-response assumed in Definition 7, after global pooling. However, we emphasize that by definition, object images and types are entirely determined by the detector, as images that give a delta response in a single output feature map. All images that result in a delta response in an output feature map, $j$ are said to be object images of type $j$, regardless of their class affiliation. Hence, it is only for trained models that the object image types will tend to coincide with semantically meaningful ground-truth classes.

Another interpretation of object images is as "parts of a whole", where it is assumed that image motifs consist of a collection of object images. An image of a car, for instance, will be composed of object images with type "wheel", "car body", *etc.*

We note that for self-supervised models and for models where we remove layers from the network before computing representations, the correspondence between ground-truth classes and object images becomes less straightforward. However, we can still assume that the model is trained to recognize *something*. The outputs of intermediate layers will therefore still correspond to objects or features that are related to what the model is trained to recognize.

We will now define multiple objects images (MO-images) as images composed of one or more object images. This definition gives rise to a natural notion of object "count" in the image, which is necessary to formalize the NCH.

**Definition 8** (Multiple objects image). An MO-image, $I \in \mathcal{I}_{C,W,H}$, constructed from object images in $\Omega_0, \ldots, \Omega_{C'-1}$, w.r.t. the detector $f$, is defined as

$$I = \sum_{j=0}^{C'-1} \sum_{(O,x',y') \in \mathcal{P}_j} \text{Tr}_{x',y'}(O). \tag{13}$$

where $C'$ is the number of object types. $\mathcal{P}_j$ is a set of 3-tuples, where the first element is an object image from $\Omega_j$, and the second third elements are the positions of that object image in $I$. $\mathcal{P}_j$ can also be empty, indicating that $I$ contains no object image of type $j$.

We now have the following theorem stating that the NCH is true for detectors and global pooling operators applied to MO-images.

**Theorem 1** (Norm-count hypothesis – simplified setting). *Let $f : \mathcal{I}_{C,W,H} \to \mathcal{I}_{C',W',H'}$ be a relaxed detector with object image types $\Omega_0, \ldots, \Omega_{C'-1}$, and let $I \in \mathcal{I}_{C,W,H}$ be a MO-image constructed from the same object images. Then, if $\mathbf{z} = [z_0, \ldots, z_{C'-1}]^\top \in \mathbb{R}^{C'}$ is the output of a global pooling operator applied to the feature maps $f(I)$, we have*

$$|z_k| = |\operatorname{Pool}(f(I))_k| \leq \gamma_k \left(|\mathcal{P}_k| + WHn_\mathcal{P}|a_{f,k}|\right), \quad k \in \mathbb{N}^0_{<C'} \tag{14}$$

*for non-negative numbers $\gamma_0, \ldots, \gamma_{C-1}$ independent of $I$.*

Theorem 1 shows that the $k$-th component is an increasing affine transformation of the number of object images of type $k$ in $I$. Furthermore, if the detector $f$ has $A_f = 0$ we get exact proportionality between $|z_k|$ and $|\mathcal{P}_k|$.

Taking the norm of $\mathbf{z}$ with elements bounded as in Equation (14) gives the following corollary.

**Corollary 1.1** ($L_p$ norm of $\mathbf{z}$). *For $p > 0$, the $L_p$ norm of $\mathbf{z}$ from Theorem 1 is*

$$||\mathbf{z}||_p \leq \left( \sum_{k=0}^{C'-1} \gamma_k^p \left(|\mathcal{P}_k| + WHn_\mathcal{P}|a_{f,k}|\right)^p \right)^{\frac{1}{p}} \tag{15}$$

Corollary 1.1 shows that the $L_p$ norm of representations is upper bounded by a monotonically increasing function of the count, corroborating the NCH.

**Corollary 1.2** (Strict detector and GAP). *If $f$ is a strict detector, and GAP is used in place of Pool, Theorem 1 simplifies to*

$$z_k = \operatorname{GAP}(f(I))_k = \frac{|\mathcal{P}_k|}{W'H'} + n_\mathcal{P}a_k \tag{16}$$

*where $n_\mathcal{P} = \sum_{j=0}^{C'-1} |\mathcal{P}_j| - 1$.*

Corollary 1.2 shows that for strict detectors followed by GAP, there is an affine relationship between each component of $\mathbf{z}$, and the number of objects of the corresponding type present in the image. Furthermore, if we have $a_k = 0$, we have exact proportionality between $z_k$ and $|\mathcal{P}_k|$. Since linear CNNs are strict detectors with $a_k = 0$, this corollary proves that each dimension, in a representation produced by linear CNNs, is proportional to the number of object images in the given MO-image.

Furthermore, Theorem 1 states that the norm of $\mathbf{z}$ is directly related to the absolute (total) count of objects in the image, regardless of the type of the object images. This is expected, since the perfect detector produces a set of delta-responses, and the global pooling operator aggregates these over the spatial dimensions.

### 3.1 Semantic information in angles

In contrast to the norm, the angle of $\mathbf{z}$ depends on the count of one object type relative to the count of another object type. This is demonstrated by the following result.

**Result 1** (Semantic information in angles). *Suppose $I_1, I_2 \in \mathcal{I}_{C,W,H}$ are MO-images processed by a strict detector $f$ with $a_k = 0$, followed by GAP. Furthermore, assume that $I_1$ only consists of objects of type $j$, and that $I_2$ only consists of objects of type $k$. This gives*

$$\mathbf{z}_1 = \mathrm{GAP}(f(I_1)) = \frac{|\mathcal{P}_j^{(1)}|}{W'H'}\mathbf{e}_j \quad and \quad \mathbf{z}_2 = \mathrm{GAP}(f(I_2)) = \frac{|\mathcal{P}_k^{(2)}|}{W'H'}\mathbf{e}_k \tag{17}$$

*where $\mathbf{e}_j$ ($\mathbf{e}_k$) denotes the vector where element $j$ ($k$) is $1$, and all other elements are $0$.*
*Then, if $(\|\mathbf{z}\|, \boldsymbol{\theta}(\mathbf{z}))$ denotes the transformation of $\mathbf{z}$ to hyperspherical coordinates, we can consider the following two cases:*

1. *Different class, same count: $j \neq k$ and $|\mathcal{P}_j^{(1)}| = |\mathcal{P}_k^{(2)}|$, which gives*

$$\|\mathbf{z}_1\| = \|\mathbf{z}_2\| \quad and \quad \boldsymbol{\theta}(\mathbf{z}_1) \neq \boldsymbol{\theta}(\mathbf{z}_2) \tag{18}$$

2. *Same class, different count: $j = k$ and $|\mathcal{P}_j^{(1)}| \neq |\mathcal{P}_k^{(2)}|$, which gives*

$$\|\mathbf{z}_1\| \neq \|\mathbf{z}_2\| \quad and \quad \boldsymbol{\theta}(\mathbf{z}_1) = \boldsymbol{\theta}(\mathbf{z}_2) \tag{19}$$

In both cases in Result 1, the angles $\boldsymbol{\theta}(\mathbf{z}_1)$ and $\boldsymbol{\theta}(\mathbf{z}_2)$ are most informative of the image classes (object types). When the images belong to different classes (case 1), the discriminative power lies in the angles and not in the norms. Conversely, when $I_1$ and $I_2$ belong to the same class (case 2), the within class distance is 0 for the angles, but non-zero for the norms. The angle thus encodes information about which classes (object types) that were detected in the image – *i.e.* the semantic information.

### 3.2 Feature norm vs. object size

It might be natural to think that the norm of a representation is positively correlated with the size of the object in the image, since larger object should lead to more, and possibly stronger activations. However, CNNs are not size equivariant, meaning that this is not necessarily the case. This is because convolutions – the basic building blocks of CNNs – detect patterns with a certain size. Resizing the patterns by contracting or dilating spatial dimensions can therefore completely change the response, both reducing og increasing its strength. In the context of our work, this means that object images are not resizeable: If one resizes an object image, it may no longer be an object image for the same detector, as it might no longer give a delta response. This, however depends on the properties of the detector, and whether it has been trained to produce similar responses for objects with different sizes. Section 4.2.2 includes an experimental analysis of the relationship between object size and feature norm.

## 4 Experiments

The purpose of these experiments is to experimentally investigate the NCH in a controlled setting. We design the experiments to have fine-grained control of the "count" in each image. This allows us to properly examine the relationship between norm and count – both quantitatively and qualitatively. Our experiments are conducted with both supervised and self-supervised models on several datasets.

Although our theoretical results hold for arbitrary $L_p$ norms, we focus on $L_2$ norms in the experimental evaluation. This is because the $L_2$ norm is the one most frequently encountered in other works (see Section 2), and has known benefits related to optimization (Liu et al., 2021), as well as alignment, uniformity, and class separability (Wang & Isola, 2020).

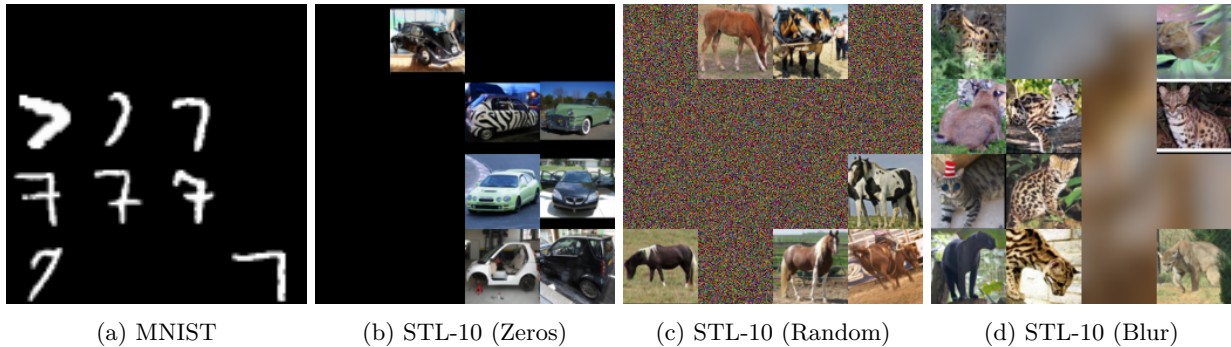

(a) MNIST      (b) STL-10 (Zeros)      (c) STL-10 (Random)      (d) STL-10 (Blur)

Figure 1: Example synthetic images generated from MNIST and STL-10, with different filling approaches for the latter.

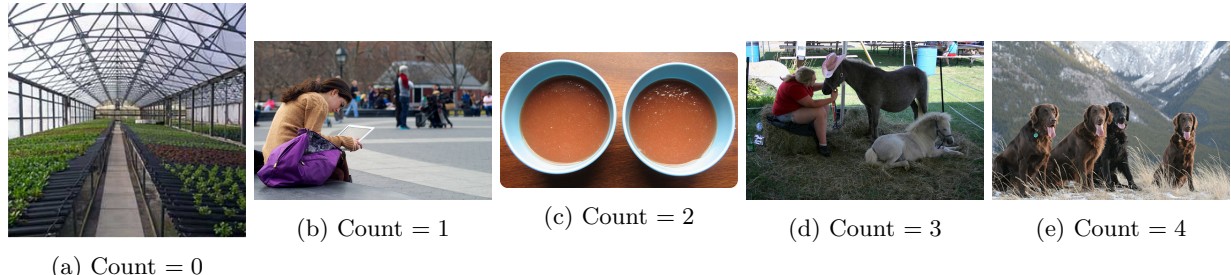

(b) Count = 1      (c) Count = 2      (d) Count = 3      (e) Count = 4

(a) Count = 0

Figure 2: Example images from the MSO dataset.

### 4.1 Setup

#### 4.1.1 Datasets

**Synthetic datasets.** In order to mimic the properties of MO-images in evaluation, we start with datasets consisting of natural images (MNIST (Lecun et al., 1998) and STL-10 (Coates et al., 2011)). Then, to generate a single evaluation image, we sample a random number of images, and place them at random positions in a $4 \times 4$ grid. This gives us an image that resembles an MO-image, where we know the true count – *i.e.* the number of object images.

For MNIST, we fill the empty grid positions with 0-values, as indicated by Definition 8. For STL-10, we generate datasets with 3 different approaches to filling the empty grid slots:

- *Zeros*: empty grid positions are filled with 0-values.

- *Random*: empty grid positions are filled with random Gaussian noise with the same mean and standard deviation as the object images.

- *Blur*: the background for the whole grid is a random blurred image, and the object images are placed on top of this image.

We experiment with different fill modes to ensure that the results are not skewed by changes in global image statistics, such as mean and variance. Figure 1 shows examples of the generated images.

**Real-world counting and object detection datasets.** In addition to the syntetic datasets, we include two real-world datasets in our experimental evaluation:

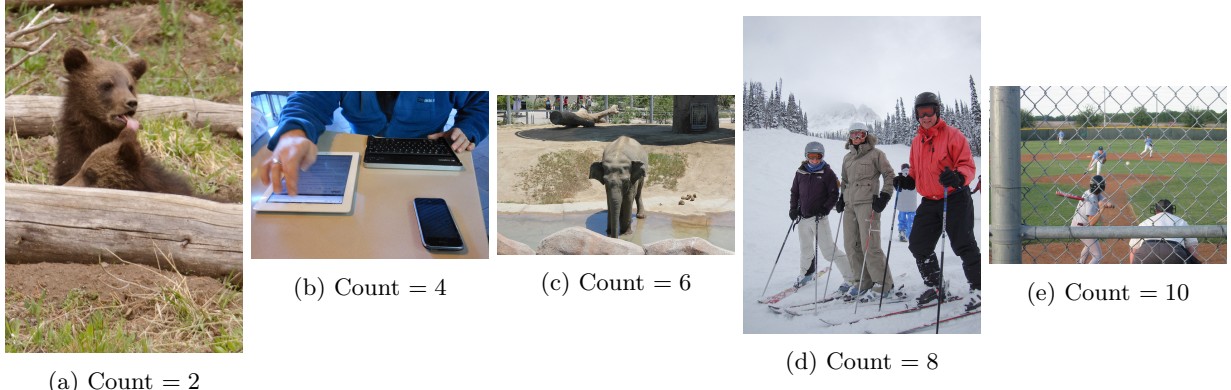

(a) Count = 2

(b) Count = 4

(c) Count = 6

(d) Count = 8

(e) Count = 10

Figure 3: COCO example images.

1. The multi salient objects (MSO) dataset[1] (Figure 2) is derived from the Salient Object Subitizing dataset (Zhang et al., 2015), and includes images with a varying number of salient foreground objects. Each image in the dataset is annotated with a count between 0 and 4 objects.

2. The common objects in context (COCO) dataset (Lin et al., 2014) (Figure 3) is a large-scale object detection dataset with images containing multiple objects from several classes. We use the number of ground-truth bounding boxes for an image as the object count. In order to ensure that each count has a representative number of images, we select images with $2 \leq count \leq 9$.

As can be seen in Figures 2 and 3, MSO contains objects that are more prominent in the image, while COCO contains objects that might be more difficult to detect, with greater variation in size and clarity.

### 4.1.2 Models and architectures

**Architectures.** Our evaluation is performed with models using the following two CNN architectures:

- `Simple-6`: A simple 6-layer CNN followed by GAP. The model has ReLU activations, max pooling after every second convolutional layer. No batch-normalization or other forms of normalization is applied anywhere in the architecture.

- `ResNet-50`: The standard 50-layer residual network architecture by He et al. (2016), with batch normalization.

For both models we add a final convolutional layer, followed by GAP to produce the final representation.

**Supervised models.** The supervised models are trained using the standard cross-entropy loss, with the softmaxed final representation (GAP output) as the model's prediction. We thus avoid any additional fully-connected classification layers, making the model more likely to satisfy the detector assumptions given in Definitions 4 and 5.

The `Simple-6` architecture is trained on MNIST, starting from randomly initialized weights. The `ResNet-50` architecture is initialized with pre-trained weights from ImageNet training[2], and fine-tuned on single-object images from COCO.

**Self-supervised models.** We include two different self-supervised learning models in our evaluation:

---

[1] `https://www.kaggle.com/datasets/jessicali9530/mso-dataset`

[2] Weights: `IMAGENET1K_V2`, documentation: `https://pytorch.org/vision/main/models/generated/torchvision.models.resnet50.html#torchvision.models.resnet50`

- The standard SimCLR model (Chen et al., 2020), with a two-layer projection head, and a normalized temperature-scaled cross entropy loss. The input to the projection head is the final representation (GAP output). Similar to the supervised setup, we train the `Simple-6` architecture on MNIST from randomly initialized weights. The `ResNet-50` architecture is initialized with self-supervised pre-trained ImageNet weights[3], and fine-tuned on single-object images from COCO.

- The Dense Contrastive Learning (DenseCL) model (Wang et al., 2021) with a `ResNet-50` encoder. This is a self-supervised model aimed to provide better representations for pixel-level predictions and multi-label data. For this model we use the pre-trained weights directly without fine-tuning. We do this both to assess our hypothesis with a model that has not been fine-tuned on single-object images, and to avoid the large computational demands required to train DenseCL.

### 4.1.3 Implementation

The experiments are implemented in Python with the PyTorch framework (Paszke et al., 2019). We will make the code for our experiments publicly available upon publication of the paper.

### 4.1.4 Quantitative evaluation of monotonic increase

We use a weighted linear regression model to quantitatively assess whether there is an increasing relationship between Z-score normalized feature norms, $\nu_i$, and the count, $c(\mathbf{x}_i)$

$$\nu_i = \beta_0 + \beta_1 c(\mathbf{x}_i) + \epsilon_i \tag{20}$$

where

$$\nu_i = \frac{||\mathbf{z}_i|| - \mu_{||\mathbf{z}||}}{\sigma_{||\mathbf{z}||}}, \quad \mu_{||\mathbf{z}||} = \frac{1}{n}\sum_{i=1}^{n}||\mathbf{z}_i||, \quad \sigma_{||\mathbf{z}||} = \sqrt{\frac{1}{n-1}\sum_{i=1}^{n}\left(||\mathbf{z}_i|| - \mu_{||\mathbf{z}||}\right)^2}, \tag{21}$$

and the residual, $\epsilon_i$, is assumed to be Gaussian with zero mean and standard deviation $\sigma_{c(\mathbf{x}_i)}$. We allow the standard deviation to be count-dependent to account for heteroskedasticity in the data.

The parameter estimates $(\hat{\beta}_0, \hat{\beta}_1)$ are computed using weighted least squares. Based on these estimates, we can test for monotonic increase by checking whether the slope $\beta_1$ is positive:

$$H_0 : \beta_1 \leq 0 \quad \text{vs.} \quad H_1 : \beta_1 > 0. \tag{22}$$

We report the estimated slope $\hat{\beta}_1$ and the $p$-value for the above hypothesis test.

## 4.2 Results

### 4.2.1 Relationship between norm and count

The results in Table 1 show a monotonic increase between norm and count for almost all experimental configurations. According to these results, the expected value of a representation norm increases by roughly 0.1 standard deviations per additional object in the image.

Our results show that the relationship between norm and count is more prominent for the synthetic grid-datasets, compared to the datasets composed of natural, real-world images (Figure 4). This is expected, since the synthetic datasets include clear, single-object images on top of a non-informative background with little distraction. We also observe a difference between the real-world datasets, with a stronger relationship between norm and count on MSO compared to COCO. This is likely caused by the more prominent objects in MSO, compared to the more complex scenes in COCO, with the latter including partially occluded, smaller, and more diverse objects.

Table 1: Estimated slopes ($\hat{\beta}_1$) and $p$-values for the linear regression model for norm vs. count. A positive slope with a low $p$-value indicates a statistically significant, monotonically increasing relationship between norm and count.

| Dataset | Model | Slope ($\hat{\beta}_1$) | $p$-value |
|---|---|---|---|
| MNIST (Zeros) | Supervised (`Simple-6`) | 0.209 | 0.000 |
| | SimCLR (`Simple-6`) | 0.126 | 0.000 |
| STL-10 (Zeros) | Supervised (`ResNet-50`) | 0.087 | 0.000 |
| | SimCLR (`ResNet-50`) | 0.134 | 0.000 |
| STL-10 (Random) | Supervised (`ResNet-50`) | -0.101 | 1.000 |
| | SimCLR (`ResNet-50`) | 0.106 | 0.000 |
| STL-10 (Blur) | Supervised (`ResNet-50`) | 0.133 | 0.000 |
| | SimCLR (`ResNet-50`) | 0.075 | 0.000 |
| MSO | Supervised (`ResNet-50`) | 0.040 | 0.032 |
| | SimCLR (`ResNet-50`) | 0.175 | 0.000 |
| | DenseCL (`ResNet-50`) | 0.474 | 0.000 |
| COCO | Supervised (`ResNet-50`) | 0.004 | 0.000 |
| | SimCLR (`ResNet-50`) | 0.009 | 0.000 |
| | DenseCL (`ResNet-50`) | -0.028 | 1.000 |

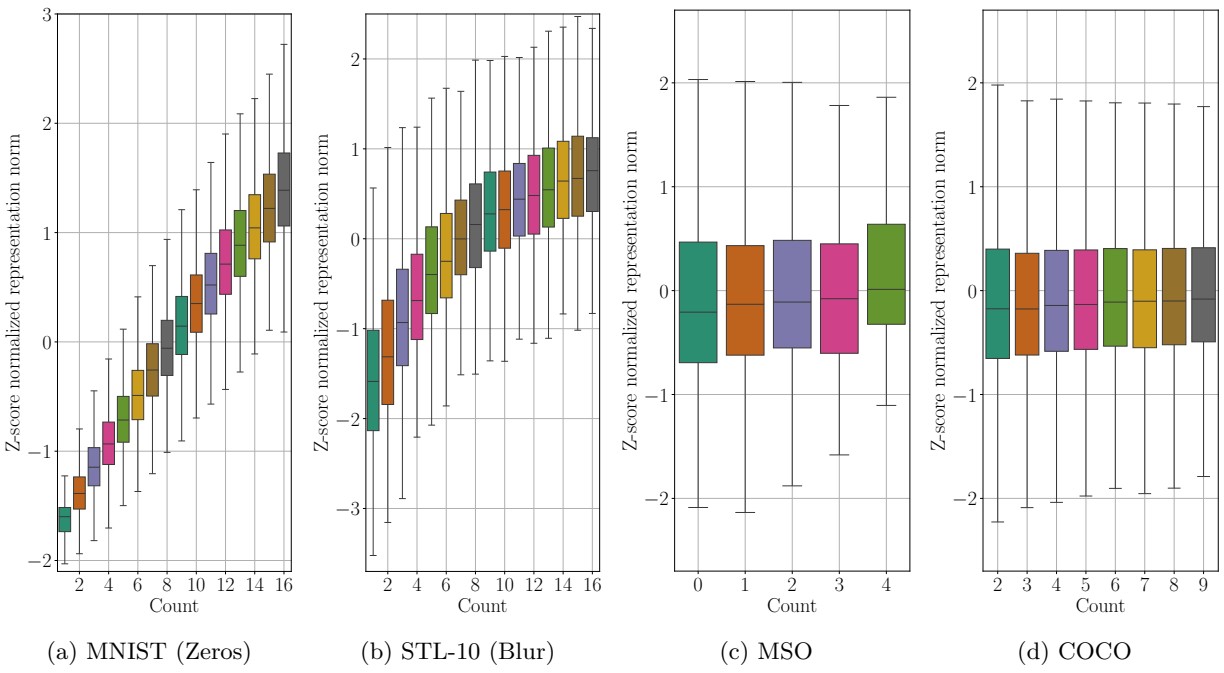

(a) MNIST (Zeros)  (b) STL-10 (Blur)  (c) MSO  (d) COCO

Figure 4: Boxplots illustrating the relationship between norm and count for supervised models.

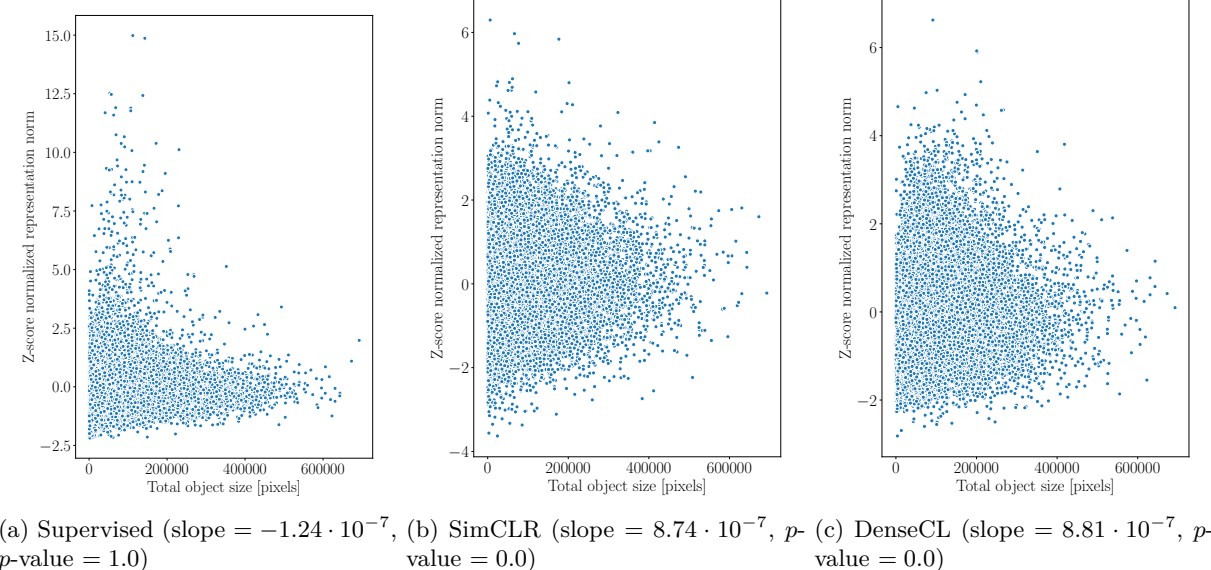

(a) Supervised (slope $= -1.24 \cdot 10^{-7}$, (b) SimCLR (slope $= 8.74 \cdot 10^{-7}$, $p$- (c) DenseCL (slope $= 8.81 \cdot 10^{-7}$, $p$-
$p$-value $= 1.0$)         value $= 0.0$)         value $= 0.0$)

Figure 5: Relationship between norm and object size on COCO. Slopes and $p$-values are from a linear regression model for norm vs. object size, with a one-sided test for positive slopes.

### 4.2.2 Relationship between norm and object size.

In order to experimentally investigate our arguments about norm object size from Section 3.2, we evaluate the relationship between norm and object size on COCO. We chose the COCO dataset for these experiments since it is a real-world dataset with bounding box annotations, allowing us to infer the size of each object as the area of the bounding box. We repeat the norm-size experiments for both the supervised and self-supervised models.

The plots in Figure 5 show no clear relationship between norm and object size. Regressing norm against count shows a small but *negative* correspondence between the two for the supervised model, and very small positive slopes ($< 10^{-6}$) for the self-supervised models. These negative and almost-zero slopes show that the correspondence between norm and size is significantly weaker compared to the correspondence between norm and count, thus corroborating our arguments in Section 3.2.

## 5 Conclusion and future work

We have presented the norm-count hypothesis (NCH) – providing novel insight in the norms of representations produced by CNNs. Under certain assumptions on the model and input images, we proved the NCH, showing that each component in a given representation is upper-bounded by the number of object images present in the input image. Moreover, from our theoretical analysis, it follows that representation norms carry information related to count, whereas angles represent semantic information. Our experiments with both supervised and self-supervised models, applied on synthetic and real-world datasets, show increasing trends between norm and count on the majority of experimental configurations - corroborating the NCH.

We believe that understanding representation norms through object counts is a promising direction of research. Although we focus on CNNs in this work, our results might generalize to other architectures and learning regimes. With vision transformers (ViTs) recently showing remarkable performance in computer vision, it should be investigated whether there is a relationship between norm and count for ViT architectures as well. It is however not trivial to prove that ViT components meet the detector conditions. We therefore leave further analyses of ViTs and the NCH to future work.

---

[3]`https://pytorch-lightning-bolts.readthedocs.io/en/latest/models/self_supervised.html#imagenet-baseline-for-simclr`

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

## A    Proofs

### A.1    Proposition 1

*Proof.*

1. Invoking condition (4) of strict detectors gives

$$(g \circ f)(I_1 + I_2) = g(f(I_1 + I_2)) = g(f(I_1) + f(I_2) + A_f) \tag{23}$$
$$= g(f(I_1)) + g(f(I_2)) + g(A_f) + 2A_g \tag{24}$$
$$= (g \circ f)(I_1) + (g \circ f)(I_2) + g(A_f) + 2A_g \tag{25}$$

2. By conditions (4) and (5), we have

$$|(g \circ f)(I_1 + I_2)| = |g(f(I_1 + I_2))| = |g(f(I_1) + f(I_2) + A_f)| \tag{26}$$
$$\leq |g(f(I_1))| + |g(f(I_2))| + |g(A_f)| + 2|A_g| \tag{27}$$
$$= |(g \circ f)(I_1)| + |(g \circ f)(I_2)| + |g(A_f)| + 2|A_g| \tag{28}$$

$\square$

### A.2    Proposition 2

*Proof.* The proposition follows directly from convolutions being linear and translation equivariant. See *e.g.* (Jähne, 2002, Ch. 4). $\square$

### A.3 Proposition 3

*Proof.* For images $I_1$ and $I_2$ we have

$$\text{BN}_{\mathbf{b},\mathbf{g},\boldsymbol{\mu},\boldsymbol{\sigma}}(I_1 + I_2)(k,x,y) = \frac{I_1(k,x,y) + I_2(k,x,y) - \mu_k}{\sigma_k} g_k + b_k \tag{29}$$

$$= \frac{g_k}{\sigma_k}(I_1(k,x,y) + I_2(k,x,y)) + b_k - \frac{\mu_g g_k}{\sigma_k} \tag{30}$$

$$= \left(\frac{g_k}{\sigma_k}I_1(k,x,y) + b_k - \frac{\mu_g g_k}{\sigma_k}\right) + \left(\frac{g_k}{\sigma_k}I_2(k,x,y) + b_k - \frac{\mu_g g_k}{\sigma_k}\right) - \left(b_k - \frac{\mu_k g_k}{\sigma_k}\right) \tag{31}$$

$$= \text{BN}_{\mathbf{b},\mathbf{g},\boldsymbol{\mu},\boldsymbol{\sigma}}(I_1)(k,x,y) + \text{BN}_{\mathbf{b},\mathbf{g},\boldsymbol{\mu},\boldsymbol{\sigma}}(I_2)(k,x,y) + \left(\frac{\mu_k g_k}{\sigma_k} - b_k\right) \tag{32}$$

$\square$

### A.4 Proposition 4

*Proof.* Let $a_1 = I_1(k,x,y)$ and $a_2 = I_2(k,x,y)$. Since addition is commutative, we can assume $a_1 \geq a_2$ without loss of generality.

Observe that, if $a_1$ and $a_2$ is positive (negative), then $a_1 + a_2$ will be positive (negative). This means that $\text{LeakyReLU}_\alpha(a_1 + a_2) = \text{LeakyReLU}_\alpha(a_1) + \text{LeakyReLU}_\alpha(a_2)$ in this case.

On the other hand, if $a_2 < 0 < a_1$, we have $|\text{LeakyReLU}_\alpha(a_1)| + |\text{LeakyReLU}_\alpha(a_2)| = |a_1| + \alpha|a_2|$. Now we consider the following two cases:

1. If $|a_1| \geq |a_2|$, then

$$|\text{LeakyReLU}_\alpha(a_1 + a_2)| = |a_1 + a_2| \tag{33}$$

$$\leq |a_1| \tag{34}$$

$$\leq |a_1| + \alpha|a_2| \tag{35}$$

$$= |\text{LeakyReLU}_\alpha(a_1)| + |\text{LeakyReLU}_\alpha(a_2)| \tag{36}$$

2. If $|a_1| \leq |a_2|$, then

$$|\text{LeakyReLU}_\alpha(a_1 + a_2)| = \alpha|a_1 + a_2| \tag{37}$$

$$\leq \alpha|a_2| \tag{38}$$

$$\leq |a_1| + \alpha|a_2| \tag{39}$$

$$= |\text{LeakyReLU}_\alpha(a_1)| + |\text{LeakyReLU}_\alpha(a_2)| \tag{40}$$

$\square$

### A.5 Proposition 5

*Proof.* Setting $\frac{1}{WH} = \gamma_k$ gives

$$GAP(I)_k = \gamma_k \sum_{x=0}^{W-1} \sum_{y=0}^{H-1} I(k,x,y), \quad k \in \mathbb{N}_{<C}^0 \tag{41}$$

from which it follows that

$$|GAP(I)_k| \leq \gamma_k \left| \sum_{x=0}^{W-1} \sum_{y=0}^{H-1} I(k,x,y) \right|, \quad k \in \mathbb{N}_{<C}^0 \tag{42}$$

$\square$

### A.6 Theorem 1

*Proof.* Since $f$ is a relaxed detector, we have

$$|f(I)| = \left| f\left( \sum_{j=0}^{C'-1} \sum_{(O,x',y')\in\mathcal{P}_j} \text{Tr}_{x',y'}(O) \right) \right| \tag{43}$$

$$\preccurlyeq \sum_{j=0}^{C'-1} \sum_{(O,x',y')\in\mathcal{P}_j} |f(\text{Tr}_{x',y'}(O))| + n_\mathcal{P}|A_f| \tag{44}$$

where $n_\mathcal{P} = \sum_{j=0}^{C'-1} |\mathcal{P}_j| - 1$.

Then, since $f$ is translation equivariant, and provides delta detections

$$|f(I)| \preccurlyeq \sum_{j=0}^{C'-1} \sum_{(O,x',y')\in\mathcal{P}_j} |f(\text{Tr}_{x',y'}(O))| + n_\mathcal{P}|A_f| \tag{45}$$

$$= \sum_{j=0}^{C'-1} \sum_{(O,x',y')\in\mathcal{P}_j} \delta_{(j,x',y')} + n_\mathcal{P}|A_f|. \tag{46}$$

Applying a global pooling operator to $f(I)$ then gives

$$|z_k| = |\text{Pool}(f(I))_k| \le \gamma_k \left| \sum_{x=0}^{W'-1} \sum_{y=0}^{H'-1} f(I)(k,x,y) \right| \tag{47}$$

$$= \gamma_k \sum_{x=0}^{W'-1} \sum_{y=0}^{H'-1} \left( \sum_{j=0}^{C'-1} \sum_{(O,x',y')\in\mathcal{P}_j} \delta_{(j,x',y')}(k,x,y) + n_\mathcal{P}|a_{f,k}| \right) \tag{48}$$

$$= \gamma_k \sum_{x=0}^{W'-1} \sum_{y=0}^{H'-1} \left( \sum_{(O,x',y')\in\mathcal{P}_k} \delta_{(k,x',y')}(k,x,y) + n_\mathcal{P}|a_{f,k}| \right) \tag{49}$$

$$= \gamma_k \left( \sum_{(O,x',y')\in\mathcal{P}_k} \delta_{(k,x',y')}(k,x',y') + WHn_\mathcal{P}|a_{f,k}| \right) \tag{50}$$

$$= \gamma_k \left( |\mathcal{P}_k| + WHn_\mathcal{P}|a_{f,k}| \right) \tag{51}$$

$\square$

### A.7 Corollary 1.1

*Proof.* The $L_p$ norm of $\mathbf{z}$ is defined as

$$||\mathbf{z}||_p = \left( \sum_{k=0}^{C'-1} |z_k|^p \right)^{\frac{1}{p}} \tag{52}$$

for $p > 0$. Since each $|z_k|$ is positive and upper bounded by $\gamma_k|\mathcal{P}_k|$ (by Theorem 1), we have

$$\sum_{k=0}^{C'-1} \gamma_k|\mathcal{P}_k|^p \ge \sum_{k=0}^{C'-1} |z_k|^p \tag{53}$$

which gives

$$\left( \sum_{k=0}^{C'-1} \gamma_k|\mathcal{P}_k|^p \right)^{\frac{1}{p}} \ge \left( \sum_{k=0}^{C'-1} |z_k|^p \right)^{\frac{1}{p}} = ||\mathbf{z}||_p \tag{54}$$

□

### A.8 Corollary 1.2

*Proof.* This proof follows the same steps as the proof of Theorem 1, but without absolute values, and with equality instead of inequality.

Since $f$ is a strict detector, we have

$$f(I) = f\left(\sum_{j=0}^{C'-1} \sum_{(O,x',y')\in\mathcal{P}_j} \mathrm{Tr}_{x',y'}(O)\right) \tag{55}$$

$$= n_\mathcal{P} A_f + \sum_{j=0}^{C'-1} \sum_{(O,x',y')\in\mathcal{P}_j} (f(\mathrm{Tr}_{x',y'}(O))) \tag{56}$$

$$= n_\mathcal{P} A_f + \sum_{j=0}^{C'-1} \sum_{(O,x',y')\in\mathcal{P}_j} f(\mathrm{Tr}_{x',y'}(O)) \tag{57}$$

where $n_\mathcal{P} = \sum_{j=0}^{C'-1} |\mathcal{P}_j| - 1$. Then, since $f$ is translation equivariant, and provides delta detections

$$f(I) = n_\mathcal{P} A_f + \sum_{j=0}^{C'-1} \sum_{(O,x',y')\in\mathcal{P}_j} f(\mathrm{Tr}_{x',y'}(O)) = n_\mathcal{P} A_f + \sum_{j=0}^{C'-1} \sum_{(O,x',y')\in\mathcal{P}_j} \delta_{(j,x',y')}. \tag{58}$$

Applying a global pooling operator to $f(I)$ then gives

$$z_k = \mathrm{GAP}(f(I))_k = \frac{1}{W'H'} \sum_{x=0}^{W'-1} \sum_{y=0}^{H'-1} f(I)(k,x,y) \tag{59}$$

$$= \frac{1}{W'H'} \sum_{x=0}^{W'-1} \sum_{y=0}^{H'-1} \left(n_\mathcal{P} A_f(k,x,y) + \sum_{j=0}^{C'-1} \sum_{(O,x',y')\in\mathcal{P}_j} \delta_{(j,x',y')}(k,x,y)\right) \tag{60}$$

$$= \frac{1}{W'H'} \sum_{x=0}^{W'-1} \sum_{y=0}^{H'-1} \left(n_\mathcal{P} a_{f,k} + \sum_{(O,x',y')\in\mathcal{P}_k} \delta_{(k,x',y')}(k,x,y)\right) \tag{61}$$

$$= \frac{1}{W'H'} \sum_{(O,x',y')\in\mathcal{P}_k} \delta_{(k,x',y')}(k,x',y') + n_\mathcal{P} a_{f,k} \tag{62}$$

$$= \frac{1}{W'H'} |\mathcal{P}_k| + n_\mathcal{P} a_{f,k} \tag{63}$$

□

## B  Experiments with overlapping object images

Table 2: Quantitative regression results for MNIST when digits are randomly placed and allowed to overlap.

| Dataset | Model | Slope ($\hat{\beta}_1$) | $p$-value |
|---------|-------|-------------------------|-----------|
| MNIST (overlapping digits) | Supervised (`Simple-6`) | 0.302 | 0.000 |
| | SimCLR (`Simple-6`) | 0.018 | 0.000 |

Table 2 and Figure 6 show the norm-count correspondence on a variant of MNIST where a number of digits are placed randomly in a $128 \times 128$ image. The results show that there is still a strong trend between norm and count, event though the digit images are allowed to overlap with each other.

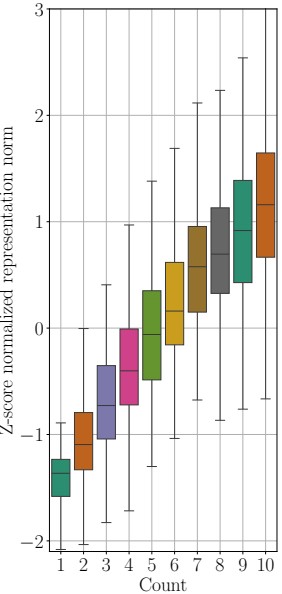 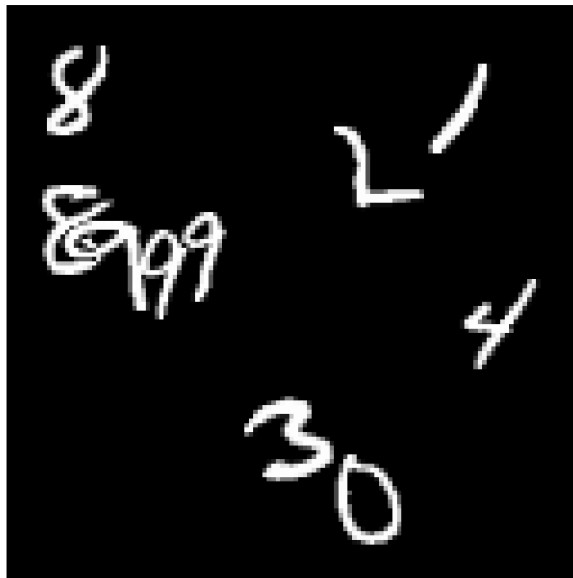

Figure 6: Left: Correspondence between norm and count when objects are allowed to overlap. Right: Example image with overlapping digits.

## C    Stepwise evaluation of monotonicity

In order to have a more fine-grained evaluation of the monotonicity in the relationship between norm and count, we measure the mean change in norm when increasing the count by 1. The results are reported in Table 3, and shows that the norm increases in the majority of cases ($> 70\%$). We observe the same trend as in Table 1 where the increase is stronger for the synthetic dataset, and is weakest for COCO, which is the most difficult dataset.

We emphasize that for almost all experimental configurations, the *negative changes are smaller, on average, compared to the positive changes.* This means that we have an overall positive trend between norm and count, as found in Section 4.2.

Table 3: Change in mean $Z$-score-normalized norm when increasing the count by 1.

| Dataset | Model | $0 \to 1$ | $1 \to 2$ | $2 \to 3$ | $3 \to 4$ | $4 \to 5$ | $5 \to 6$ | $6 \to 7$ | $7 \to 8$ |
|---|---|---|---|---|---|---|---|---|---|
| MNIST (Zeros) | Supervised (`Simple-6`) | – | 1.040 | 1.001 | 0.830 | 0.697 | 0.630 | 0.686 | 0.499 |
| | SimCLR (`Simple-6`) | – | -0.828 | -0.219 | 0.110 | 0.173 | 0.527 | 0.548 | 0.576 |
| STL10 (Zeros) | Supervised (`ResNet-50`) | – | 0.201 | 0.281 | 0.255 | 0.226 | 0.205 | 0.197 | 0.056 |
| | SimCLR (`ResNet-50`) | – | 1.111 | 0.524 | 0.330 | 0.143 | 0.048 | 0.005 | 0.009 |
| STL10 (Random) | Supervised (`ResNet-50`) | – | 0.029 | -0.070 | -0.086 | -0.010 | 0.006 | -0.222 | -0.018 |
| | SimCLR (`ResNet-50`) | – | 0.990 | 0.477 | 0.352 | 0.184 | 0.080 | 0.137 | 0.025 |
| STL10 (Blur) | Supervised (`ResNet-50`) | – | 0.352 | 0.445 | 0.234 | 0.364 | 0.247 | 0.304 | 0.191 |
| | SimCLR (`ResNet-50`) | – | 0.639 | 0.341 | 0.213 | 0.112 | 0.119 | 0.041 | -0.065 |
| MSO | Supervised (`ResNet-50`) | 0.030 | 0.025 | 0.047 | 0.166 | – | – | – | – |
| | SimCLR (`ResNet-50`) | 0.461 | -0.075 | 0.317 | -0.415 | – | – | – | – |
| | DenseCL (`ResNet-50`) | 1.026 | -0.004 | 0.297 | 0.003 | – | – | – | – |
| COCO | Supervised (`ResNet-50`) | – | – | -0.006 | 0.011 | -0.025 | 0.035 | -0.007 | 0.089 |
| | SimCLR (`ResNet-50`) | – | – | 0.065 | 0.028 | 0.023 | -0.000 | -0.029 | -0.013 |
| | DenseCL (`ResNet-50`) | – | – | -0.035 | 0.003 | -0.055 | -0.028 | -0.059 | 0.055 |

| Dataset | Model | $8 \to 9$ | $9 \to 10$ | $10 \to 11$ | $11 \to 12$ | $12 \to 13$ | $13 \to 14$ | $14 \to 15$ | $15 \to 16$ |
|---|---|---|---|---|---|---|---|---|---|
| MNIST (Zeros) | Supervised (`Simple-6`) | 0.520 | 0.550 | 0.485 | 0.406 | 0.383 | 0.347 | 0.453 | 0.338 |
| | SimCLR (`Simple-6`) | 0.606 | 0.575 | 0.574 | 0.683 | 0.619 | 0.632 | 0.635 | 0.573 |
| STL10 (Zeros) | Supervised (`ResNet-50`) | 0.159 | -0.062 | 0.095 | -0.007 | 0.037 | 0.029 | -0.003 | 0.008 |
| | SimCLR (`ResNet-50`) | -0.047 | -0.016 | -0.004 | 0.113 | 0.113 | 0.165 | 0.453 | 0.561 |
| STL10 (Random) | Supervised (`ResNet-50`) | -0.066 | -0.033 | -0.119 | -0.167 | -0.167 | -0.318 | -0.300 | -0.589 |
| | SimCLR (`ResNet-50`) | -0.008 | -0.050 | -0.100 | -0.031 | -0.050 | 0.087 | 0.136 | 0.559 |
| STL10 (Blur) | Supervised (`ResNet-50`) | 0.212 | 0.017 | 0.134 | 0.126 | 0.105 | 0.151 | 0.076 | 0.057 |
| | SimCLR (`ResNet-50`) | 0.008 | 0.058 | -0.072 | 0.071 | -0.049 | 0.163 | 0.169 | 0.128 |
| MSO | Supervised (`ResNet-50`) | – | – | – | – | – | – | – | – |
| | SimCLR (`ResNet-50`) | – | – | – | – | – | – | – | – |
| | DenseCL (`ResNet-50`) | – | – | – | – | – | – | – | – |
| COCO | Supervised (`ResNet-50`) | -0.058 | – | – | – | – | – | – | – |
| | SimCLR (`ResNet-50`) | -0.020 | – | – | – | – | – | – | – |
| | DenseCL (`ResNet-50`) | -0.059 | – | – | – | – | – | – | – |

