# OpenReview forum: "Norm-count Hypothesis: On the Relationship Between Norm and Object Count in Visual Representations"
_TMLR — Rejected by TMLR_

### Review · Reviewer_9nhV · 2024-06-11

**Summary Of Contributions:**

This paper proposed the Norm-Count Hypothesis, which states a monotonic relationship between the number of objects in the input image and the norm of its embedding CNN outputs. Theoretically, it is shown that when the input image consists of multiple objects and the model is the composition of relaxed detectors and a global pooling operator, the embedding norm is upper-bounded by the weight p-norm-like quantity determined by the number of objects. In the case of the combination of strict detectors and global average pooling, there is a linear relationship between these two quantities. In addition, it is shown that convolution layers and batch normalization without learnable parameters are strict detectors and that Leaky ReLU is a relaxed detector. For the numerical analysis, the hypothesis is tested by applying supervised and self-supervised learning models to process real image datasets (MNIST, STL-10) and object detection datasets (MSO, COCO).

**Audience:**

Yes

**Broader Impact Concerns:**

No broader impact concerns

**Claims And Evidence:**

No

**Requested Changes:**

- P3, Definition 4: The definition of strict and relaxed detectors is clear. However, the motivation and validity of adopting this definition are not clear. Also, an intuitive explanation of the definition and theorem of strict and relaxed detectors would increase the validity of this definition.
- P3, Definition 2: Notational inconsistency: $\mathrm{Tr}$ (P3, Definition 2) and $T$ (P5, Definition 7).
- P5, Definition 7: According to Definition 6, the set of all object images of type $j$ (i.e., $\Omega_j$) depends on the detector $f$. Therefore, the detector $f$ must also be explicit in Definition 7.
- P5, Theorem 1, eq. (14): $|\mathrm{Pool}(I)|_k$ -> $|\mathrm{Pool}(f(I))|_k$
- P6, Corollary 1.2, eq. (14): $|\mathrm{GAP}(I)|_k$ -> $|\mathrm{GAP}(f(I))|_k$
- P10: $\mu_{\|\boldsymbol{z}\|}$ and $\sigma_{\|\boldsymbol{z}\|}$ are not used.

**Strengths And Weaknesses:**

*Strengths*

- By introducing the concept of strict and relaxed detectors, the formulation of NCH is formalized without restricting to specific architectures (e.g., CNN, BN).
- The NCH is theoretically shown for realistic architectures (that is, linear CNN with batch normalization)

*Weaknesses*

- The definition of Leaky ReLU and the proof that it is a relaxed detector (Proposition 4) is questionable
- Even if Proposition 4 is correct, NCH is not directly demonstrated theoretically because Theorem 1 is just the upper bounds of the norm of an embedding when Leaky ReLU activation functions are included.
- It is not easy to judge that the numerical experimental results are affirmative to NCH (especially Figure 4 (c), (d)).


*Questions*

- In the definition of Leaky ReLU (equation (8)), $\alpha$ is set to the value within $[0, 1)$. However, to our knowledge, $\alpha$ is usually a negative value. For example, the default value of PyTorch implementation is $\alpha=-0.01$ [1].
- Even if we adopt this paper's definition of Leaky ReLU, the proof of Proposition 4 needs to be corrected. Specifically, in Equation (32), the inequality $|a_1 + a_2|\leq |a_1 + \alpha a_2|$ for $a_2 < 0 < a_1$, is not correct. For example, $a_1 = 1, a_2 = -2, \alpha=0.5$ would be a counterexample.
- Section 4.2.1 tests the positivity of $\hat{\beta}$ and claims that NCH is valid on the ground that $\hat{\beta}> 0$ is supported in many cases. However, it is verified whether the fitting to a linear function (or other monotonically increasing function) is appropriate in the first place. If the error term $\epsilon_i$ is large, it is not known whether $\nu_i$ and $c(\boldsymbol{x}_i)$ has monotonic relationship (related to the next comment)
- We see in Figure 4 (c) and (d) the high variance of embedding norms, questioning the validity of NCH.

[1] https://pytorch.org/docs/stable/generated/torch.nn.LeakyReLU.html

---

> ### Author Response · Authors · 2024-07-08
> **Review response**
>
> We thank the Reviewer for a thorough review of our work. Detailed answers to comments and questions are given below.
>
> ---
>
> __Comment__:
>
> * _The definition of Leaky ReLU and the proof that it is a relaxed detector (Proposition 4) is questionable_
> * _In the definition of Leaky ReLU (equation (8)), $\alpha$ is set to the value within $[0, 1)$. However, to our knowledge, $\alpha$ is usually a negative value. For example, the default value of PyTorch implementation is $\alpha=-0.01$ [1]._
> * _Even if we adopt this paper's definition of Leaky ReLU, the proof of Proposition 4 needs to be corrected. Specifically, in Equation (32), the inequality $|a_1 + a_2|\leq |a_1 + \alpha a_2|$ for $a_2 < 0 < a_1$, is not correct. For example, $a_1 = 1, a_2 = -2, \alpha=0.5$ would be a counterexample._
>
>
> __Response__:
>
> We follow the original definition of LeakyReLU, where $\alpha$ is a small positive number (Section 2, https://arxiv.org/pdf/1505.00853). The parameter name `negative_slope` in the PyTorch documentation referenced by the Reviewer, likely refers to the slope for negative inputs and not that the slope itself should be negative. This is also reflected by the (positive) default value: $\alpha$ = `negative_slope` = `0.001`.
>
> We thank the reviewer for identifying the error in the proof of Proposition 4. We have made a minor adjustment to the definition of relaxed detectors, and updated the proof of Proposition 4. This should now be correct.
>
> ---
>
> __Comment__:
>
> * _Even if Proposition 4 is correct, NCH is not directly demonstrated theoretically because Theorem 1 is just the upper bounds of the norm of an embedding when Leaky ReLU activation functions are included._
>
> __Response__:
>
> In Corollary 1.2 we show that there is exact equality when we have strict detectors and global average pooling. Regarding Theorem 1, it is not straightforward to determine the tightness of the bound, since this depends on the tightness of the bounds in Definitions 5 and 6. Tighter bounds in Definitions 5 and 6 would lead to a tighter bound in Theorem 1.
>
> ---
>
> __Comment__:
>
> * *Section 4.2.1 tests the positivity of $\hat{\beta}$ and claims that NCH is valid on the ground that $\hat{\beta}> 0$ is supported in many cases. However, it is verified whether the fitting to a linear function (or other monotonically increasing function) is appropriate in the first place. If the error term $\epsilon_i$ is large, it is not known whether $\nu_i$ and $c(\boldsymbol{x}_i)$ has monotonic relationship (related to the next comment)*
>
> __Response__:
>
> The linear model is valid as long as the relationship between norm and count is linear, regardless of large variance. If the relationship was zero or negative, this would cause the linear model to have a zero or negative slope. We have verified that the norm-count relationship does not significantly violate the linearity assumption, indicating that the linear model is valid.
>
> ---
>
> __Comment__:
>
> * _It is not easy to judge that the numerical experimental results are affirmative to NCH (especially Figure 4 (c), (d))._
> * _We see in Figure 4 (c) and (d) the high variance of embedding norms, questioning the validity of NCH._
>
> __Response__:
>
> Although the variance is large for MSO and COCO, the p-values in Table 1 indicate a monotonically increasing relationship for almost all configurations.
>
> We hypothesise that the large variance is likely caused by the following two factors:
> 1. The CNNs are not perfect detectors, meaning that the ground-truth classes do not lead to perfect delta-like responses in the output feature maps.
> 2. In some images, there is ambiguity in the object count. Hence, there might be a difference between "actual count" and the "count indicated by the number of objects belonging to a ground-truth class". This effect is illustrated in Figure 3 (c)-(e), where it is not easy to find all the objects that contribute to the overall count.
>
> ---
>
> __Comment__:
>
> * _P3, Definition 4: The definition of strict and relaxed detectors is clear. However, the motivation and validity of adopting this definition are not clear. Also, an intuitive explanation of the definition and theorem of strict and relaxed detectors would increase the validity of this definition._
>
> __Response__:
>
> We have updated the manuscript with additional motivation and explanation of the definitions of strict and relaxed detectors.
>
> ---
>
> __Comment__:
>
> (Other requested changes)
>
> __Response__:
>
> We thank the reviewer for identifying notational inconsistencies and errors. We have corrected these in the revised manuscript.
>
> ---

---

> > ### Comment · Reviewer_9nhV · 2024-07-17
> > **Response to authors' comments**
> >
> > I thank the authors for responding to my review comments. Here is their reply.
> >
> > > We follow the original definition of LeakyReLU, where $\alpha$ is a small positive number (Section 2, https://arxiv.org/pdf/1505.00853). The parameter name negative_slope in the PyTorch documentation referenced by the Reviewer, likely refers to the slope for negative inputs and not that the slope itself should be negative. This is also reflected by the (positive) default value: $\alpha$ = negative_slope = 0.001.
> >
> > I am sorry that it was my mistake. The author is correct, and I would like to retract this.
> >
> > --------------------
> > > We thank the reviewer for identifying the error in the proof of Proposition 4. We have made a minor adjustment to the definition of relaxed detectors, and updated the proof of Proposition 4. This should now be correct.
> >
> > I thank the authors for updating the proof. The updated version looks correct to me.
> >
> > --------------------
> >
> > > In Corollary 1.2 we show that there is exact equality when we have strict detectors and global average pooling.
> > > Regarding Theorem 1, it is not straightforward to determine the tightness of the bound, since this depends on the tightness of the bounds in Definitions 5 and 6. Tighter bounds in Definitions 5 and 6 would lead to a tighter bound in Theorem 1.
> >
> > First, in my initial review comments, I misunderstood NCH and thought that it claims the linear relationship between object count and norm. However, Definition 1 just claims a monotonic relationship between the two. I apologize for my mistake.
> > Nevertheless, although I understand that it is difficult to show equality, I think that the fact that only the upper bound is shown for the relaxed operators, including Leaky ReLU, weakens the argument for the establishment of NCH.
> >
> > --------------------
> >
> > > The linear model is valid as long as the relationship between norm and count is linear, regardless of large variance. If the relationship was zero or negative, this would cause the linear model to have a zero or negative slope. We have verified that the norm-count relationship does not significantly violate the linearity assumption, indicating that the linear model is valid.
> >
> > > Although the variance is large for MSO and COCO, the p-values in Table 1 indicate a monotonically increasing relationship for almost all configurations.
> >
> > In my understanding, the hypothetical tests implicitly assume that the data can be modeled with a monotonic function, i.e., monotonicity between object count $x$ and norm $y$, and test the positivity of coefficients.  It means that the fact that the hypothesis $\beta > 0$ is adopted does not imply the existence of a monotonically increasing relationship.
> > For example, suppose we have a quadratic relation $y=(x-2)^2$ for $1\geq x\geq 10$. The hypothesis $\beta > 0$ would still be adopted because of an overall positive trend. However, there is no monotonically increasing relationship between $x$ and $y$ as claimed by NCH.
> > This example suggests that we need to verify the monotonic relationship by means other than the positive-coefficient test.
> >
> > --------------------
> >
> > > We have updated the manuscript with additional motivation and explanation of the definitions of strict and relaxed detectors.
> >
> > OK
> >
> > --------------------
> >
> > > We thank the reviewer for identifying notational inconsistencies and errors. We have corrected these in the revised manuscript.
> >
> > OK

---

> > > ### Author Response · Authors · 2024-07-18
> > > **Response to Reviewer's comments**
> > >
> > > >First, in my initial review comments, I misunderstood NCH and thought that it claims the linear relationship between object count and norm. However, Definition 1 just claims a monotonic relationship between the two. I apologize for my mistake. Nevertheless, although I understand that it is difficult to show equality, I think that the fact that only the upper bound is shown for the relaxed operators, including Leaky ReLU, weakens the argument for the establishment of NCH.
> > >
> > > The upper bound in Theorem 1 is a direct result of the inequalities in Definitions 5 and 6. We completely agree that the result would be stronger if we could prove equality instead of an upper bound, but this is not possible with the current definitions (without restricting the analysis to linear CNNs). Changing the definitions to include equality would be a highly relevant direction for future follow-up work on the NCH.
> > >
> > > >In my understanding, the hypothetical tests implicitly assume that the data can be modeled with a monotonic function, i.e., monotonicity between object count $x$ and norm $y$, and test the positivity of coefficients. It means that the fact that the hypothesis $\beta>0$ is adopted does not imply the existence of a monotonically increasing relationship. For example, suppose we have a quadratic relation $𝑦=(𝑥−2)^2$  for $1 \le x \le 10$. The hypothesis $\beta>0$ would still be adopted because of an overall positive trend. However, there is no monotonically increasing relationship between $x$ and $y$ as claimed by NCH. This example suggests that we need to verify the monotonic relationship by means other than the positive-coefficient test.
> > >
> > > We agree that there can exist edge-cases, such as the one pointed out here, where the linear evaluation procedure is unable to capture sub-regions of non-monotonicity. We have however, not observed many such cases in our experiments, and it is not trivial to define a metric that accurately captures these cases.
> > >
> > > As an alternative evaluation procedure, Appendix C in the revised manuscript includes a table showing changes in average norm when the count is increased by one. This table offers a more fine-grained evaluation of the relationship between norm and count. We observe the same trend as in the main results section, with a monotonic relationship for the synthetic datasets, and a weaker, mostly monotonic relationship for MSO and COCO.
> > >
> > > We emphasize that for almost all experimental configurations, the *negative changes are smaller, on average, compared to the positive changes.* This means that we have an overall positive trend between norm and count.

---

### Review · Reviewer_bML5 · 2024-06-21

**Summary Of Contributions:**

As pointed out in the introduction of the paper, the problem of proper representation learning is long standing and one of the main approaches is to apply $L_2$ normalization. There are many works which pointed out that such normalization simplifies optimization problems and provides strong representation learning. In this paper, the authors go further and formulate the hypothesis that the representation's norm reflects the number of objects in the image while the angle reflects the semantic relationship between objects on different images. Authors also pose that for most of the vision tasks we don't need information about the number of the objects, thus this information is redundant and we need to be invariant even to it. The latter leads to $L_2$ normalized representations which outperform not-normalized. Authors provide theoretical analysis of their hypothesis in the limited setting of CNNs with batchnorm, leaky relu and global pooling (with some relaxation on some operations) and then validate their theory via empirical analysis for CNNs (simple CNN and resnet-50) with synthetic images (images with one object, e.g. MNIST, are stacked to form bigger image) or real images (e.g. COCO). While for synthetic images there is a clear trend on the dependence between the norm and number of objects, for real images (COCO), there is mostly no correlation which authors attribute to the complexity of the images.

**Audience:**

Yes

**Broader Impact Concerns:**

Overall the work is theoretical with general empirical results on the images of any content, thus no concerns with the current paper for the ethical implications.

**Claims And Evidence:**

No

**Requested Changes:**

Text is written overall very well. Only minor suggestions to improve readability:
- "a monotonically increasing relationship" - this is used several times in text and even shown in empirical analysis that this relationship is in place. But I still don't get what you mean by "monotonically increasing relationship" - it is increasing with the growth of number of objects in the image? Maybe it is good to define exactly the functional dependence you want to see and its properties.
- sec 2.1 ending: "little work exists on this topic" - not clear which exact topic you mean here.
- sec 2.2 "hyperspherical embeddings" -> here all embeddings usage should be weights, as you speak about network weights normalization
- def 4: $a_k$ what is index $k$ here?
- proposition 2: I didn't get why after conv operator the image size is bigger by the size of kernel - 1. What padding do you assume here? Often we use the same pad in the models, how will this change the theory then? (I think all will be valid, but maybe good to have discussion on the different padding for conv operator that all remains the same).
- definition 7 - maybe add note that $P_j$ can be an empty set? so that we allow that image to have less number of objects or even zero of a particular type.
- type on page 7 "reducing og increasing" -> "reducing or increasing"
- A.8 after (47) why is there "-1" in $n_P$ definition?

Other questions / requests:
- I think def 6 is only theoretical and in practice this property will not hold for the real images, so that on every base object it acts like a delta function. We can imagine that we have some basis in the image space on which the detector is giving us the delta function, but there is no empirical analysis in the paper to show some correspondence here between the basis and the exact images. Thus, right now it is a very simplified definition. One can refer here to the neural collapse phenomenon, and say the centroids will be these basis images, but it was shown later that neural collapse is very model / regularization dependent and there is mismatch between train / test regimes. Thus I could not find a good explanation on the assumption made for the theorem 1 in general so that it is really practical.
- any checks that the object images are the real objects? even for synthetic data?
- Theorem 1 states that if there is no object type in the image then the $z_k=0$. I don't see this check in the empirical analysis, also it is a really impractical property, I don't think I ever saw in the representations that some coordinates are zeros. I can imagine that the theorem gives structure of the space, so that norm corresponds to number of objects and we can find the basis where we can have $z_k=#objects$ of type $k$, but then we often have non-linear space mapping anyway in NNs and we use this highly non-linear space - thus not clear how this property (norm reflects number of objects) is preserved in that non-linear space (I believe it could not be preserved). I feel there is a missing piece with the invariance property that norm property is preserved with the non-linear mapping (which could be not true and then theory is kind of helpless for the general NNs we use nowadays).
- what happens in the synthetic data experiments if you don't stack objects but do sum of the images? e.g. for MNIST, so that objects will intersect with each other? Does the theorem still hold?
- what happens if we have softmax operation? or even switch to transformer models / attention-based? Nowadays they are shown to have better scaling properties. What happens if we have a bigger model / other variants of resnet?
- Statement in the end of the sec 5 on extending theory is overclaimed - I don't see how you can simply extend theory for transformers e.g.
- What happens with other domains? e.g. speech and text, is it counting the words for the sentence embeddings e.g.?
- It is not clear why for detection problems we also remove norm dependence, as here it should be useful really.

**Strengths And Weaknesses:**

I have read carefully all proofs and theoretical analysis as well as the rest of the paper.

**Strengths**
- Idea on the norm hypothesis is very reasonable and makes sense. It is also interesting, will be interesting to share with a broader audience and can have a significant impact on understanding of representation learning in future.
- Theory part is good: notation is well defined, simple to follow, and I didn't find any errors or typos. Also it is easy to parse math, which is good and rare for theoretical papers nowadays. Thanks for careful writing.
- Attempt to include synthetic data experiments and real data experiments to show that theoretical results are aligned with practice

**Weaknesses**
- There is no analysis / extension for the other type of data, like text or speech, which really limits the hypothesis right now as representation learning is used for many different domains. E.g. analysis for sentence embeddings will be interesting to check if it counts specific words / n-grams in the sentence which reflects the norm and maybe we wanna have the norm.
- Synthetic data analysis should include results when we have an image as a sum of images and not the stack, e.g. allowing intersection of objects - this will be close to COCO and thus we can understand better how to transfer to COCO.
- Results for real data are very weak and not convincing: from Table 1 and Fig 4 I would say that the norm hypothesis does not hold for real images at all.
- I think some analysis disentanglement from the optimization itself should be taken into account, e.g. what exactly and how we use the learned representations.
- Empirical results only for very limited architectures, without extension to transformers / attention operation e.g. I presume weak results even for resnet on real data already show that is it not generalizable from synthetic data, thus I expect it will not hold for ViT models and other SSL methods.
- For theory I believe it is very limited with assumption on the basis property and kind of linearity of the space, which will not hold in more complicated NNs with more non-linearities, thus highly non-linear space cannot be expressed with the basis of object images which is doing the trick on theorem proving.

---

> ### Author Response · Authors · 2024-07-09
> **Review response (Part 1)**
>
> We thank the Reviewer for a thorough review of our work, and for finding our work to be interesting and theoretically sound. Detailed answers to comments and questions are given below.
>
> ---
>
> __Comment__:
>
> * _There is no analysis / extension for the other type of data, like text or speech, which really limits the hypothesis right now as representation learning is used for many different domains. E.g. analysis for sentence embeddings will be interesting to check if it counts specific words / n-grams in the sentence which reflects the norm and maybe we wanna have the norm._
> * _What happens with other domains? e.g. speech and text, is it counting the words for the sentence embeddings e.g.?_
>
> __Response__:
>
> The theoretical analysis will hold for any data that is input to a 1-D or 2-D CNN. These are common architectures in e.g. acoustics and speech analysis.
>
> We focus on images and visual representations in this work, since $L_2$ normalization is extremely common in several branches of computer vision, such as supervised, self-supervised and few-shot learning. However, we agree that it would be interesting to investigate the applicability of the NCH in other areas and for other architectures in future work.
>
> ---
>
> __Comment__:
>
> * _Synthetic data analysis should include results when we have an image as a sum of images and not the stack, e.g. allowing intersection of objects - this will be close to COCO and thus we can understand better how to transfer to COCO._
> * _what happens in the synthetic data experiments if you don't stack objects but do sum of the images? e.g. for MNIST, so that objects will intersect with each other? Does the theorem still hold?_
>
> __Response__:
>
> We have included a new experiment where MNIST digits are randomly placed in the image, and allowed to overlap with each other.
> The results are included in Appendix B in the revised manuscript, and show that there still is a strong correspondence between norm and count.
>
> We observe that the variance of norms is higher in the experiments where digits are allowed to overlap. As the Reviewer writes, it is possible that this explains some of the large norm variance observed with the COCO dataset, as this dataset also contains overlapping objects.
>
> ---
>
> __Comment__:
>
> * _Results for real data are very weak and not convincing: from Table 1 and Fig 4 I would say that the norm hypothesis does not hold for real images at all._
>
> __Response__:
>
> Although the variance is large for MSO and COCO, the p-values in Table 1 indicate a monotonically increasing relationship for almost all configurations.
>
> We hypothesise that the large variance is likely caused by the following two factors:
> 1. The CNNs are not perfect detectors, meaning that the ground-truth classes do not lead to perfect delta-like responses in the output feature maps.
> 2. In some images, there is ambiguity in the object count. Hence, there might be a difference between "actual count" and the "count indicated by the number of objects belonging to a ground-truth class". This effect is illustrated in Figure 3 (c)-(e), where it is not easy to find all the objects that contribute to the overall count.
>
> ---
>
> __Comment__:
>
> * _I think some analysis disentanglement from the optimization itself should be taken into account, e.g. what exactly and how we use the learned representations._
>
> __Response__:
>
> The dependency of the theoretical contributuions on optimization lies in whether the detector's object images coincide with ground truth classes or semantic objects. This is not an explicit dependency, but rather implicit and related to whether the model meets the requirements set by Definition 4 or 5.
>
> ---
>
> __Comment__:
>
> * _Empirical results only for very limited architectures, without extension to transformers / attention operation e.g. I presume weak results even for resnet on real data already show that is it not generalizable from synthetic data, thus I expect it will not hold for ViT models and other SSL methods._
>
> __Response__:
>
> In this work we have focused on conducting a theoretically sound analysis, with experiments that lie close to the theorectical contributions. Hence, to avoid making claims not supported by our theory about other architectures, we have limited the experiments to CNNs. It is, however, not entirely correct to presume that the NCH will not hold for ViTs, based on the current results. This would require further theoretical and experimental analysis.

---

> ### Author Response · Authors · 2024-07-09
> **Review response (Part 2)**
>
> __Comment__:
>
> * _For theory I believe it is very limited with assumption on the basis property and kind of linearity of the space, which will not hold in more complicated NNs with more non-linearities, thus highly non-linear space cannot be expressed with the basis of object images which is doing the trick on theorem proving._
> * _I think def 6 is only theoretical and in practice this property will not hold for the real images, so that on every base object it acts like a delta function. We can imagine that we have some basis in the image space on which the detector is giving us the delta function, but there is no empirical analysis in the paper to show some correspondence here between the basis and the exact images. Thus, right now it is a very simplified definition. One can refer here to the neural collapse phenomenon, and say the centroids will be these basis images, but it was shown later that neural collapse is very model / regularization dependent and there is mismatch between train / test regimes. Thus I could not find a good explanation on the assumption made for the theorem 1 in general so that it is really practical._
>
> __Response__:
>
> Due to the complexity of neural network models it is both common and necessary to make assumptions when conducting theoretical analyses. Delta-like responses is the goal of standard supervised learning, meaning that the delta-assumption is not as limiting as it might appear initially. Further, it is important to note that the "distributive over addition"-properties (Eqs. (4) and (5)) of detectors is not a linearity assumption. The theory is therefore not restricted to linear mappings only.
>
> ---
>
> __Comment__:
>
> *"a monotonically increasing relationship" - this is used several times in text and even shown in empirical analysis that this relationship is in place. But I still don't get what you mean by "monotonically increasing relationship" - it is increasing with the growth of number of objects in the image? Maybe it is good to define exactly the functional dependence you want to see and its properties.*
>
> __Response__:
>
> The statement "a monotonically increasing relationship" means that the norm is a strictly increasing function of count (i.e. the number of object images).
>
> We have added a sentence in the revised manucript, after the NCH is first introduced, clarifying what we mean by "monotonically increasing relationship".
>
> ---
>
> __Comment__:
>
> _proposition 2: I didn't get why after conv operator the image size is bigger by the size of kernel - 1. What padding do you assume here? Often we use the same pad in the models, how will this change the theory then? (I think all will be valid, but maybe good to have discussion on the different padding for conv operator that all remains the same)._
>
> __Response__:
>
> Images are assumed to be zero-padded, as indicated by Definition 1. We therefore assume "full" padding in the convolution operator. This is why the output image becomes larger. However, the theory will also hold for both "valid" and "same" padding. We thank the reviewer for pointing this out and have updated the manuscript with a brief discussion on padding strategies.
>
> ---
>
> __Comment__:
>
> _any checks that the object images are the real objects? even for synthetic data?_
>
> __Response__:
>
> The supervised models are trained to produce one-hot outputs, with the one-value in the feature map index that corresponds to the ground-truth class. Since object images are defined as those that produce a delta (one-hot) response, highly accurate models will tend to have ground-truth classes as object image types. In our experiments, model training and fine-tuning resulted in accuracies $ > 85 $ %, meaning that there likely is a significant overlap between object images and ground-truth classes.
>
> The self-supervised models are trained without label information, meaning that the correspondence between ground-truth classes and object image types is not explicitly enforced during training. However, we also observe high accuracy for these models, meaning that the object iimages recognized by the model are related to the ground-truth classes.

---

> ### Author Response · Authors · 2024-07-09
> **Review response (Part 3)**
>
> __Comment__:
>
> *Theorem 1 states that if there is no object type in the image then the z_k = 0. I don't see this check in the empirical analysis, also it is a really impractical property, I don't think I ever saw in the representations that some coordinates are zeros. I can imagine that the theorem gives structure of the space, so that norm corresponds to number of objects and we can find the basis where we can have z_k=#objects of type , but then we often have non-linear space mapping anyway in NNs and we use this highly non-linear space - thus not clear how this property (norm reflects number of objects) is preserved in that non-linear space (I believe it could not be preserved). I feel there is a missing piece with the invariance property that norm property is preserved with the non-linear mapping (which could be not true and then theory is kind of helpless for the general NNs we use nowadays).*
>
> __Response__:
>
> Many modern architectures use piecewise linear functions to create highly non-linear mappings from inputs to represenations. Proposition 4 in the manuscript shows that LeakyReLU (a widely used piecewise linear activatiopn function) is a relaxed detector. Hence, our thery also applies to non-linear mappings.
>
> ---
>
> __Comment__:
>
> _what happens if we have softmax operation? or even switch to transformer models / attention-based? Nowadays they are shown to have better scaling properties. What happens if we have a bigger model / other variants of resnet?_
>
> __Response__:
>
> In CNNs, softmax is usually applied after the representation has been computed, to produce probabilistic outputs. Because of this, the softmax function is not directly relevant to the theory presented in the manuscript. Regarding transformers, it would require further theoretical analysis to see how the softmax-operation influences representation norms.
>
> ---
>
> __Comment__:
>
> _Statement in the end of the sec 5 on extending theory is overclaimed - I don't see how you can simply extend theory for transformers e.g._
>
> __Response__:
>
> We have adjusted the claims in Section 5. The Reviewer is correct in that it is not trivial to extend the theory to e.g. vision transformers.
>
> ---
>
> __Comment__:
>
> _It is not clear why for detection problems we also remove norm dependence, as here it should be useful really._
>
> __Response__:
>
> Many classification tasks are inherently count-invariant. For instance, an image containing cats should be classified as "cat" regardless of how many cats there are in the image. However, for tasks where count is important (e.g. object detection), it might be beneficial to preserve the norm in order to keep the count information encoded in the representation.
>
> ---
>
> __Comment__:
>
> *A.8 after (47) why is there "-1" in $n_P$ definition?*
>
> __Response__:
>
> The sum
>
> $\sum\limits_{j=0}^{C'-1}\sum\limits_{(O, x', y') \in \mathcal P_j} A_f$
>
> contains $\sum\limits_{j=0}^{C'-1} |\mathcal P_j|$ terms, and thus $\sum\limits_{j=0}^{C'-1} |\mathcal P_j| - 1 = n_P$ summations. Since we have one $A_f$ per summation (Definition 4), we get
>
> $\sum\limits_{j=0}^{C'-1}\sum\limits_{(O, x', y') \in \mathcal P_j} A_f = n_P \cdot A_f$
>
> ---
>
> __Comment__:
>
> (Other requested changes.)
>
> __Response__:
>
> We thank the reviewer for identifying notational inconsistencies and errors. We have corrected these in the revised manuscript.
>
> ---

---

> > ### Comment · Reviewer_bML5 · 2024-07-13
> > **Other comments**
> >
> > Dear Authors,
> >
> > Thanks a lot for detailed answers and revision. I didn't check the revision yet (except the things you mentioned in the reply) - by any chance could be mark the changes with red color / other color so that I can quickly go over them instead of proof reading the whole paper?
> >
> > Please find below my comments on your reply (most things are resolved, though have some thoughts):
> >
> > > We have included a new experiment where MNIST digits are randomly placed in the image, and allowed to overlap with each other. The results are included in Appendix B in the revised manuscript, and show that there still is a strong correspondence between norm and count.
> > > We observe that the variance of norms is higher in the experiments where digits are allowed to overlap. As the Reviewer writes, it is possible that this explains some of the large norm variance observed with the COCO dataset, as this dataset also contains overlapping objects.
> >
> >
> > Thanks for conducting this experiment, this is now more convincing :)
> >
> > > Although the variance is large for MSO and COCO, the p-values in Table 1 indicate a monotonically increasing relationship for almost all configurations.
> >
> > Yep, true, especially with the new results with overlapping MNIST. However this pose the question if for realistic overlapping objects we have very strong presence that norm is counting objects. For me it seems that there is some truth in that but we also learn something else besides count in the norm of representations for the realistic data. Not sure how now how this is now helpful to avoid object count if norm includes other things too now. But I acknowledge that the result you got on its own is valuable and interesting.
> >
> >
> > > The dependency of the theoretical contributuions on optimization lies in whether the detector's object images coincide with ground truth classes or semantic objects. This is not an explicit dependency, but rather implicit and related to whether the model meets the requirements set by Definition 4 or 5.
> >
> > Ok
> >
> > > It is, however, not entirely correct to presume that the NCH will not hold for ViTs, based on the current results. This would require further theoretical and experimental analysis.
> >
> > Yes, agree with your point, but then it should be also said the same way in the text as I see you decided to change to (to smooth discussion about transformers as it is non-trivial extension).
> >
> > > Due to the complexity of neural network models it is both common and necessary to make assumptions when conducting theoretical analyses. Delta-like responses is the goal of standard supervised learning, meaning that the delta-assumption is not as limiting as it might appear initially. Further, it is important to note that the "distributive over addition"-properties (Eqs. (4) and (5)) of detectors is not a linearity assumption. The theory is therefore not restricted to linear mappings only.
> >
> > I think I don’t get how this delta response is considered. If this is class label - agree, but then how it is connected to the image representation (like SSL) and normalization of the representation? We never consider the final predictions for the class label as representation and moreover we on purpose exclude sometimes last layers like in SimCLR, thus you take some intermediate representation which never will be delta function (maybe could be a bit in case of neural collapse phenomenon). Maybe I still miss something here.
> >
> > > We have added a sentence in the revised manucript, after the NCH is first introduced, clarifying what we mean by "monotonically increasing relationship".
> >
> > Thanks!
> >
> > > Images are assumed to be zero-padded, as indicated by Definition 1. We therefore assume "full" padding in the convolution operator. This is why the output image becomes larger. However, the theory will also hold for both "valid" and "same" padding. We thank the reviewer for pointing this out and have updated the manuscript with a brief discussion on padding strategies.
> >
> > Thanks!
> >
> > > The self-supervised models are trained without label information, meaning that the correspondence between ground-truth classes and object image types is not explicitly enforced during training. However, we also observe high accuracy for these models, meaning that the object iimages recognized by the model are related to the ground-truth classes.
> >
> > For labeled data - ok. For SSL - not sure, as here you have extra layers to map to classes - then I expect to see kind of delta function the representation vector, not that I can map them into proper classes. I think some clarification on that object images definition is needed to remove misinterpretation this as an intermediate representation maybe.
> >
> > > We have adjusted the claims in Section 5. The Reviewer is correct in that it is not trivial to extend the theory to e.g. vision transformers.
> >
> > Thanks!

---

> > > ### Comment · Reviewer_bML5 · 2024-07-13
> > > **[continue]**
> > >
> > > > Many classification tasks are inherently count-invariant. For instance, an image containing cats should be classified as "cat" regardless of how many cats there are in the image. However, for tasks where count is important (e.g. object detection), it might be beneficial to preserve the norm in order to keep the count information encoded in the representation.
> > >
> > > Do you know of any reference where it is shown that L2 normalization is poor for object detection pre-training?
> > >
> > > > A.8 after (47) why is there "-1" in definition?
> > >
> > > I still don’t get why $\sum |P_j| - 1 = n_P$.
> > >
> > >
> > > Thanks,
> > >
> > > Reviewer.

---

> ### Author Response · Authors · 2024-07-16
> **Response to comments**
>
> Thank you for your continued interest in our work! The changes mentioned in the above responses and in the comment below have been marked with red text in the latest manuscript revision.
>
> Detailed answers to remaining comments and questions are given below.
>
> >Yep, true, especially with the new results with overlapping MNIST. However this pose the question if for realistic overlapping objects we have very strong presence that norm is counting objects. For me it seems that there is some truth in that but we also learn something else besides count in the norm of representations for the realistic data. Not sure how now how this is now helpful to avoid object count if norm includes other things too now. But I acknowledge that the result you got on its own is valuable and interesting.
>
> Thank you for finding our results valuable and interesting. We agree that there can be many factors influencing the norm -- especially for more complex images, like the ones contained in COCO. The proposed NCH contributes to understanding the norm through object count, and thus sheds light on some of the factors influencing representation norms.
>
> >Yes, agree with your point, but then it should be also said the same way in the text as I see you decided to change to (to smooth discussion about transformers as it is non-trivial extension).
>
> We agree. The final paragraph in the manuscript's conclusion (page 12) now reads:
>
> *"We believe that understanding representation norms through object counts is a promising direction of research. Although we focus on CNNs in this work, our results might generalize to other architectures and learning regimes. With vision transformers (ViTs) recently showing remarkable performance in computer vision, it should be investigated whether there is a relationship between norm and count for ViT architectures as well. It is however not trivial to prove that ViT components meet the detector conditions. We therefore leave further analyses of ViTs and the NCH to future work."*
>
> >I think I don’t get how this delta response is considered. If this is class label - agree, but then how it is connected to the image representation (like SSL) and normalization of the representation? We never consider the final predictions for the class label as representation and moreover we on purpose exclude sometimes last layers like in SimCLR, thus you take some intermediate representation which never will be delta function (maybe could be a bit in case of neural collapse phenomenon). Maybe I still miss something here.
>
> It is true that the correspondence between ground-truth classes (labels) and object images becomes less straightforward for self-supervised models, and/or when we remove layers from the model before computing representations.
>
> However, we can assume that the model is trained to recognize _something_. Because of this, the outputs of intermediate layers will still correspond to objects or features that are related to whatever the model is trained to recognize (in the case of self-supervised learning, the model is often trained to recognize objects in the image that are invariant to the chosen augmentation strategy). This makes the number of detections related to the number of objects, causing the norm to be increasing with the number of objects, as explained by the NCH.
>
> >For labeled data - ok. For SSL - not sure, as here you have extra layers to map to classes - then I expect to see kind of delta function the representation vector, not that I can map them into proper classes. I think some clarification on that object images definition is needed to remove misinterpretation this as an intermediate representation maybe.
>
> Please see the answer to the comment directly above this one. We have also updated the manuscript with a brief discussion on this topic (see page 5).
>
> >Do you know of any reference where it is shown that L2 normalization is poor for object detection pre-training?
>
> We are not aware of any such references, but we have not conducted a thorough literature search on this. Possible impacts of the NCH and $L_2$ normalization would be an interesting direction for future research.
>
> >I still don’t get why $\sum|P_j| - 1 = n_P$
>
> $n_P$ counts the number of _additions_ in the sum
>
> $$\sum\limits_{j=0}^{C'-1}\sum\limits_{(O, x', y') \in \mathcal P_j}$$
>
> this is because, from Definition 4, we get one $A_f$ for each _addition_ of two terms. The number of additions is one less than the number of terms.
>
> Consider the example of computing $f(I_1 + I_2 + I_3)$ for $k = 3$ images: $I_1, I_2, I_3$. Definition 4 gives
> $$f(I_1 + I_2 + I_3) = f(I_1) + f(I_2 + I_3) + A_f = f(I_1) + f(I_2) + f(I_3) + 2 \cdot A_f = f(I_1) + f(I_2) + f(I_3) + (k - 1) \cdot A_f $$
>
> Similarly, for arbitrary $k \ge 1$ we have
> $$ f\bigg(\sum_{i=1}^{k} I_i \bigg) = \sum_{i=1}^{k}f(I_i) + (k-1) A_f. $$
> If our explanation still is unclear, please let us know the step(s) at which the explanation is insufficient, and we will try to elaborate further.

---

> > ### Comment · Reviewer_bML5 · 2024-07-20
> > **Reply**
> >
> > Dear Authors,
> >
> > thanks for all clarifications! And I have rechecked the revised version.
> >
> > > this is because, from Definition 4 ...
> >
> > Ahh, now I got it! Thanks!
> >
> > I think all my questions / concerns resolved except maybe this delta function assumption / use (All analysis is valid, but mainly not clear how the transfer from theory to real application will be strong, though your empirical results confirm the property).
> >
> > I agree that we can assume that model learns something, but my concern is that you cannot say that representation is zero-one vector, it will be floats, thus decomposition you use with delta function and linear combination (aka vector decomposition using the basis of the vector space) will not be doable from math derivation.

---

> > > ### Author Response · Authors · 2024-07-24
> > > **Response to comment**
> > >
> > > Thank you for your continued interest in our work!
> > >
> > > >I think all my questions / concerns resolved except maybe this delta function assumption / use (All analysis is valid, but mainly not clear how the transfer from theory to real application will be strong, though your empirical results confirm the property).
> > >
> > > >I agree that we can assume that model learns something, but my concern is that you cannot say that representation is zero-one vector, it will be floats, thus decomposition you use with delta function and linear combination (aka vector decomposition using the basis of the vector space) will not be doable from math derivation.
> > >
> > > We agree that representations for real-world images will, generally, not be zero-one valued only. However, the delta assumption is necessary for the theoretical analysis, and our results show that even though the assumption is somewhat relaxed, the experimental findings are in-line with the derived theoretical results. Thus indicating that there is a correspondence between our theory and our experiments.
> > >
> > > It would certainly be interesting to investigate a relaxed delta assumption to instead incorporate arbitrary (or at least Gaussian) distributions, in the channel corresponding to the object image type. This would likely be closer to what is observed in real-world experiments. However, this is a nontrivial generalization of our current contributions, and thus we leave it to future work.

---

### Review · Reviewer_xQG5 · 2024-06-26

**Summary Of Contributions:**

Goal of the paper: Investigate the role of representation norms for images under convolutional network models.

The paper tries to prove the following simple hypothesis (named as "norm-count hypothesis"). NCH: CNNs produce image representations with higher norm for images with an increasing number of objects to be detected.

This may explain why normalizing final representations may help in count-invariant tasks (e.g. a fixed number of objects to classify, say).

**Audience:**

Yes

**Claims And Evidence:**

Yes

**Requested Changes:**

It would be interesting to study and explore similar phenomenons on VIT models.

The qualitative results of synthetic vs real-world datasets are very different (see Figure 4). Not sure if there are ways to generate more "real-looking" synthetic datasets. The confidence intervals for real-world datasets seem to be very large and they greatly overlap across object counts.

It's not quite clear to me if there are any actionable conclusions of this work, or whether this can go beyond an interesting observation.

While definitions and propositions in Section 3 are conceptually easy to follow, it's a bit of a rough read as there's no motivation or context for them.

**Strengths And Weaknesses:**

Section 3 presents the theoretical analysis. It starts by sharing a number of definitions and propositions. In particular, it shows that most of the operations present in CNNs (batch norm with fixed statistics, Leaky ReLU, convolutions) are strict or relaxed detectors. Then, it shows that if we apply a global pooling operator to any such detector, we can bound the norm of the final image representation as a function of the number of objects that form the image. In the case of strict detectors (i.e. no ReLUs) with global average pooling, Corollary 1.1 can be refined to equality. Not totally sure, as I may be missing something, but it seems the theory applies to the whole family of CNN functions, i.e. it says nothing about training dynamics and the "good" functions we'll end up with after fitting the data -- it applies to the randomly initialized CNN at the very beginning too. Also, not sure how meaningful the upper bounds are (or whether in practice the actual z-norm are much lower regardless of the count).

In Section 4, the work provides a number of experiments to empirically provide evidence of NCH. A couple of synthetic datasets are created to control the number of objects per image, and two real-world datasets are used too (COCO and MSO). Two CNN models are trained on those datasets (a simple one and a ResNet-50) on supervised and unsupervised setups. In order to study the question at hand, a linear regression is fitted using the learnt represention norms and the ground truth number of objects in each image (regress the former on the latter). In most cases, it seems the positive coefficient for number of objects is significant (using standard hypothesis testing). The paper then concludes that more objects tend to lead to higher representation norms.

Figure 4 plots results for synthetic (left) and real-world (rights) datasets.

The work also comments on the relationship between angles on images and semantic information.

---

> ### Author Response · Authors · 2024-07-08
> **Review response**
>
> We thank the Reviewer for a detailed review, and for finding our work interesting. Detailed answers to comments and questions are given below.
>
> ---
>
> __Comment__:
>
> _Not totally sure, as I may be missing something, but it seems the theory applies to the whole family of CNN functions, i.e. it says nothing about training dynamics and the "good" functions we'll end up with after fitting the data -- it applies to the randomly initialized CNN at the very beginning too._
>
> __Response__:
>
> The theory applies to all detectors. However, it is only for detectors with high precision that object images are likely to coincide with semantic ground-truth classes. I.e. the detector has to be trained to output a delta function in channel _j_ for objects with ground-truth class _j_, as is done in a standard classification setup.
>
> ---
>
> __Comment__:
>
> _Also, not sure how meaningful the upper bounds are (or whether in practice the actual z-norm are much lower regardless of the count)._
>
> __Response__:
>
> In Corollary 1.2 we show that there is exact equality when we have strict detectors and global average pooling. Regarding Theorem 1, it is not straightforward to determine the tightness of the bound, since this depends on the tightness of the bounds in Definitions 5 and 6. Tighter bounds in Definitions 5 and 6 would lead to a tighter bound in Theorem 1.
>
> ---
>
> __Comment__:
>
> _It would be interesting to study and explore similar phenomenons on VIT models._
>
> __Response__:
>
> We agree that it would be interesting to study additional architectures, such as vision transformers. It is however not trivial to show that ViT components satisfy detector conditions. We therefore leave a study of NCH and ViTs to future follow-up work.
>
> ---
>
> __Comment__:
>
> _The confidence intervals for real-world datasets seem to be very large and they greatly overlap across object counts._
>
> __Response__:
>
> Although the variance is large for MSO and COCO, the p-values in Table 1 indicate a monotonically increasing relationship for almost all configurations.
>
> We hypothesise that the large variance is likely caused by the following two factors:
> 1. The CNNs are not perfect detectors, meaning that the ground-truth classes do not lead to perfect delta-like responses in the output feature maps.
> 2. In some images, there is ambiguity in the object count. Hence, there might be a difference between "actual count" and the "count indicated by the number of objects belonging to a ground-truth class". This effect is illustrated in Figure 3 (c)-(e), where it is not easy to find all the objects that contribute to the overall count.
>
> ---
>
> __Comment__:
>
> _It's not quite clear to me if there are any actionable conclusions of this work, or whether this can go beyond an interesting observation._
>
> __Response__:
>
> L2 normalization is widely used in several areas of deep learning, such as supervised learning, self-supervised learning, and few-shot learning. Understanding what information representation norms carry can thus lead to a deeper understanding of models in all of these fields. In addition, our work can help guide the use of norms and normalization in models, based on whether the downstream task depends on object count.
>
> ---
>
> __Comment__:
>
> _While definitions and propositions in Section 3 are conceptually easy to follow, it's a bit of a rough read as there's no motivation or context for them._
>
> __Response__:
>
> We have updated the manuscript with additional motivation and explanation for several definitions in Section 3.

---

### Decision · Action_Editor_8det · 2024-07-30

**Recommendation:** Reject

**Comment:**

NCH is an interesting idea, and the work put into the paper and the authors' engagement in the discussion are appreciated.

While there is some theoretical and empirical evidence pointing in the direction of the NCH, unfortunately I believe that the evidence is still not "clear and convincing". The reviewers' recommendations are mixed, and some concerns remain following the discussion.

As one reviewer highlighted, relying on a linear regression coefficient test as the main source of evidence might be limiting. Adding predictive evidence to these experiments could increase the strength of the evidence. For example, if it were shown on two or more clean datasets of real images (such as MSO) that object count can be predicted much more accurately from representation norm alone, relative to angle, and vice versa for object class, then this would offer strong additional evidence. If such predictive experiments did not support NCH, the paper could be reframed as an exploration of what is encoded in norm and angle.

I feel that such changes are more than minor; therefore, my recommendation is rejection.

**Audience:**

I believe that analysis of learned representations is a valuable area of exploration. The hypothesis is thought-provoking, and I believe it would be of interest to TMLR's audience. The analysis is limited to batch-norm networks, and the lack of modern CNNs with LayerNorm (e.g., ConvNeXt) or Transformers does limit potential interest. The authors acknowledge this, citing the fact that the theory would be hard to extend to this case. However, given that the theory may not perfectly match real data already (see upper bound and SSL discussions above), empirical results could still be valuable.

**Claims And Evidence:**

The main idea is that there is a monotonic relationship between the representation norm and the number of certain objects in an image (the Norm Count Hypothesis, NCH). The paper presents both a theoretical and empirical analysis of this link. Both of these sources provide positive evidence for the NCH, further, some concerns from the previous submission (e.g., object size vs. count) have been resolved. However, some concerns raised by the reviewers and those raised in the previous round of reviews remain somewhat unresolved.

Empirical:

1. There is a discrepancy between synthetic and real data experiments. The empirical link weak on real data (Fig. 4). In the rebuttal, the authors cite two reasons: (i) Dataset noise. (ii) A mismatch between the theory and practice (CNNs do not exhibit the requisite "delta-like" responses as required by the theory). Regarding (i), even in the cleanest setting, a supervised model trained on MSO, the regression coefficient (regressing count to norm) is not significant. (ii) is discussed below.
2. The paper claims that if the representation norm is related to the count, then the angle corresponds to the semantics: "the angle of z depends on the count of one object type relative to the count of another object type". However, no evidence is presented for this.
3. The choice of experimental settings is not fully justified. All but one model (DenseCL) are fine-tuned on single-object COCO. This choice is not ablated.

Theoretical:

1. The theory makes use of an assumption that a trained model elicits a "delta-like" response to object images; i.e., its representation is nearly one-hot for such images, and "object images" correspond to real-world objects. However, no empirical evidence jutsifies this assumption. For supervised learning, if the representation taken is the final logits, then this property is directly encouraged by training. However, for the pre-logits (which are usually considered the "representation" layer) this will not be the case. For self-supervised learning (SSL), no theoretical guarantee exists. Despite fitting the assumption beter, it seems that self-supervised networks often yield stronger evidence for NCH. Given that the mismatch between this assumption and practice is cited as a reason for somewhat inconclusive results on real data, I feel that it is important to check that the assumption does hold in some cases, and when it does, that NCH also holds (on real data).
2. The theory introduces an upper bound that is hard to quantify; while not a critical flaw, this potentially weakens the claims.
3. One question from a reviewer, "Theorem 1 states that if there is no object type in the image then the z_k = 0. I don't see this check in the empirical analysis," seems to be unresolved.